# Simulating the Photochemical Birth of the Hydrated Electron in Liquid Water

Gonzalo Díaz Mirón [1] ✉, Cesare Malosso [2,3], Solana Di Pino [1], Colin K. Egan [4], Diganta Dasgupta[1,2], Christopher J. Mundy [5,6] & Ali Hassanali [1] ✉

The photochemistry of UV-irradiated liquid water underlies many physical, chemical, and biological processes, with the formation of the hydrated electron as a central event. Despite extensive experimental and theoretical efforts, its microscopic origin remains incompletely understood. Using excited state molecular dynamics simulations of photoexcited liquid water, we resolve the sequence of chemical events leading to hydrated electron formation on the excited state. The excitation localizes on specific hydrogen-bond network defects, followed by two competing pathways. The first produces a hydrogen atom and undergoes ultrafast non-radiative decay to the ground state within 100 femtoseconds. The other proceeds via proton-coupled electron transfer, generating hydronium ions, hydroxyl radicals, and an excited state hydrated electron. This mechanism is driven by ultrafast coupled rotational and translational motions of water molecules, forming water-mediated ion-radical pairs that persist on picosecond timescales and influence visible emission. These results provide a unified framework for interpreting time-resolved spectroscopic observations and guide future experimental and theoretical investigations.

The hydrated electron has captivated scientific interest since its discovery nearly 60 years ago[1]. This chemical species plays a fundamental role in diverse areas spanning radiation chemistry[2,3], DNA damage[4,5], and redox reactions[6,7]. One of the most widely employed methods to generate hydrated electrons involves photoexcitation across a broad range of photon energies, from UV to γ-ray irradiation[2,8,9]. This species has been generated and studied in various aqueous environments, including bulk liquids, clusters in vacuum and at interfaces[10–16]. Interestingly, the hydrated electron can be created upon photoexcitation to the first electronic excited state of liquid water[17,18]. This is particularly intriguing given that the excitation energy involved (~ 8.3 eV)[19] is well below the ionization potential of water (~ 11 eV)[19,20]. Perhaps even more surprising is that experiments have reported electron generation at photon energies as low as 6.5 eV[21,22].

Understanding how hydrated electrons are produced upon excitation to the first absorption band in liquid water requires a detailed characterization and understanding of the initial photoexcitation event. Numerous experiments have shown that excitation at different energies using either single[23,24] or two photon[17,25] within the first broad band in liquid water (~ 7–8.5 eV) open different pathways in the excited state. However, unraveling excited state dynamics after photoexcitation in liquid water remains a significant challenge due to the extremely short timescales involved (less than 1 ps) and the simultaneous formation of multiple reactive species (e.g., electrons, protons and hydroxyl radicals to name a few)[2,26,27]. While experiments can successfully identify these final products, which molecular species in the hydrogen-bond network get excited and the intermediate steps that occur immediately after

[1]Condensed Matter and Statistical Physics, The Abdus Salam International Center for Theoretical Physics, Trieste, Italy. [2]SISSA – Scuola Internazionale Superiore di Studi Avanzati, Trieste, Italy. [3]Laboratory of Computational Science and Modeling, IMX, École Polytechnique Fédérale de Lausanne, Lausanne, Switzerland. [4]Initiative for Computational Catalysis, Flatiron Institute, New York, USA. [5]Physical and Computational Sciences Directorate, Pacific Northwest National Laboratory, Richland, WA, USA. [6]Department of Chemical Engineering, University of Washington, Seattle, WA, USA.
✉e-mail: gdiaz_mi@ictp.it; ahassana@ictp.it

photoexcitation in realistic condensed phase conditions, remain largely unresolved.

To address these limitations, a variety of theoretical models and computational techniques have been developed[28]. These efforts aim to complement experimental results and provide insight into the transient processes underlying electron generation and solvation in water. A central question in this regard is whether the excitation is localized on a single water molecule or delocalized over multiple molecules. Although the dominant view supports localization[18,29,30], to the best of our knowledge, most previous theoretical studies do not capture the coupling between thermal disorder and condensed phase fluctuations, thereby limiting the generality of their conclusions. In addition, due to the quantum-mechanical nature of the hydrated electron, one conceptual and practical challenge is how to best localize its position within the hydrogen-bond network. As noted by Herbert, who has pioneered the theory and simulations of the hydrated electron in the electronic ground state, most studies rely exclusively on visual inspection of the electronic wavefunction or density which can bias the interpretation[28]. This aspect also applies to studies of underlying nature of the electronic excitation in condensed liquid water. Importantly, the hydrogen bonding environment surrounding the molecules that are excited, is known to significantly influence both the energy and intensity of electronic transitions[31–33].

Photoionization of water is known to create the solvated electron[25,34–36]. The equilibrium structure of the hydrated electron remains a subject of ongoing debate. Early theoretical models proposed either a cavity model, where the electron resides in a vacuum-like pocket surrounded by waters[37–39] or non-cavity models in which no excluded volume is formed and the electron diffuses to several water molecules[40,41]. These earlier methods relied on parameterized interactions between water molecules and the electron, which limit their predictive power. As a result, more accurate quantum chemistry approaches have become popular in recent years. Ab initio methods have been applied to water and halide-water clusters in vacuum[42–44], but how and if these results can be extrapolated to bulk water is not straightforward. Notably, some of these models failed to accurately predict the coordination number of the hydrated electron in the liquid phase[28]. Ab initio bulk water simulations have been also performed for the hydrated electron[25,34,45]. While all of these approaches have provided valuable insights into the properties of the hydrated electron at equilibrium in the electronic ground state, they do not capture the initial stages of its formation. This is due to the fact that in most of these studies, the simulations start with a pre-inserted electron in the electronic ground state and therefore do not capture the events on the excited state.

In this work, we investigate the excited state dynamics of neat liquid water following excitation to the first electronic excited state using the Restricted Open Kohn Sham (ROKS) approach[46,47] a technique that has been successfully used to investigate dissociative and charge transfer excitations in closed-shell systems[48–51]. In addition, we perform an extensive validation of this method in the context of water photochemistry, benchmarking its performance against different electronic structure settings and alternative excited state approaches. By performing simulations on an ensemble of thermally sampled configurations from the electronic ground state, we elucidate new insights into the photophysics and photochemistry of water paying special attention to the important role of fluctuations of the topology of the water hydrogen-bond network. Our study begins by analyzing the fundamental properties of the first excited state in liquid water, addressing the long-debated question of localization. The canonical manner of tackling this in quantum-chemistry is through visual inspection of the electronic wavefunction or electronic density. Here, using the Inverse Participation Ratio (IPR)[52,53] which has been previously applied to study the extent of delocalization of exciton wavefunctions in molecular systems[54], we directly probe the

partitioning of the electronic spin density among the photoexcited water molecules. In addition, the analysis of the local environment of the excited water molecules on the red-tail range of the spectrum reveal that most of them reside on topological defects in the hydrogen-bond network[29,31,55] where the asymmetry of the hydrogen bonding also plays a critical role.

Our theoretical predictions provide a coherent and unified understanding of the mechanisms underlying the photochemistry of water drawn from several different experimental time-dependent spectroscopic measurements over the last two decades. Specifically, our simulations provide new insights into the two proposed mechanisms invoked to rationalize the excited state dynamics of water, namely Hydrogen Atom Transfer (HAT) and Proton Coupled Electron Transfer (PCET). For both HAT and PCET, one fundamental challenge has been identifying which transient species form after absorption of a photon on the excited state itself, in contrast to being formed as a consequence of photodeactivation to the ground state. Our approach overcomes these hurdles predicting that the HAT mechanism exhibits a higher quantum yield, consistent with experimental observations[17]. Additionally, we observe the formation of a transient hydronium radical, a species previously proposed but not yet experimentally confirmed[56,57]. Importantly, our results also show that this mechanism does not lead to the formation of a hydrated electron in the excited state. On the other hand, for the PCET pathway, we observe the formation and early-time evolution of the hydrated electron before the system relaxes to the ground state. The creation of the hydrated electron in the excited state is in turn driven by coupled translational and rotational motions that lead to hydronium ion and hydroxyl radical pairs mediated by solvent as recently suggested by ultrafast electron diffraction measurements[35]. Finally, we demonstrate that the photon emission by the hydrated electron measured by Tauber et al.[58] is strongly modulated by the extent of the electron localization. These results offer new perspectives into tuning the color of fluorescence emission of excited state hydrated electrons in aqueous systems.

## Results
### Initial photo-absorption in liquid water
Understanding the nature of the initial photo-absorption in liquid water is a critical step toward elucidating the mechanisms that lead to hydrated electron formation, especially at excitation energies within the first absorption band. Figure 1 summarizes our main findings regarding the characteristics of this initial excitation process employing 100 different conformations sampled from well converged deep neural-network based ground state simulations (see "Methods" section for details). Figure 1A presents the experimental absorption spectrum compared with the probability density function (PDF) of the excitation energy for the $S_0 \rightarrow S_1$ transition, determined using the ROKS method for all the conformations (see "Methods" section). Our simulations predict an average excitation energy of 8.1 eV, which is in excellent agreement with the experimental value of 8.3 eV[18,19] and also with Time-Dependent Density Functional Theory (TDDFT) (see Supplementary Note 1).

To quantify the degree of the localization of the first electronic excitation, we partitioned the electronic spin density across individual water molecules of the system and computed the Inverse Participation Ratio[59] (IPR). This approach provides an agnostic manner of determining the extent to which the excitation is delocalized (see "Methods" section). As shown in the Fig. 1B, the majority of configurations involve a single water molecule in the $S_0 \rightarrow S_1$ transition, although a non insignificant fraction exhibits a more delocalized excitation spread over up to five water molecules. These events typically involve a chain of connected water molecules forming a water-wire[60–62], similar to what was recently observed in electronic structure calculations in ice[33,63]. The typical analysis used to characterize the extent of delocalization rely on visual inspection of

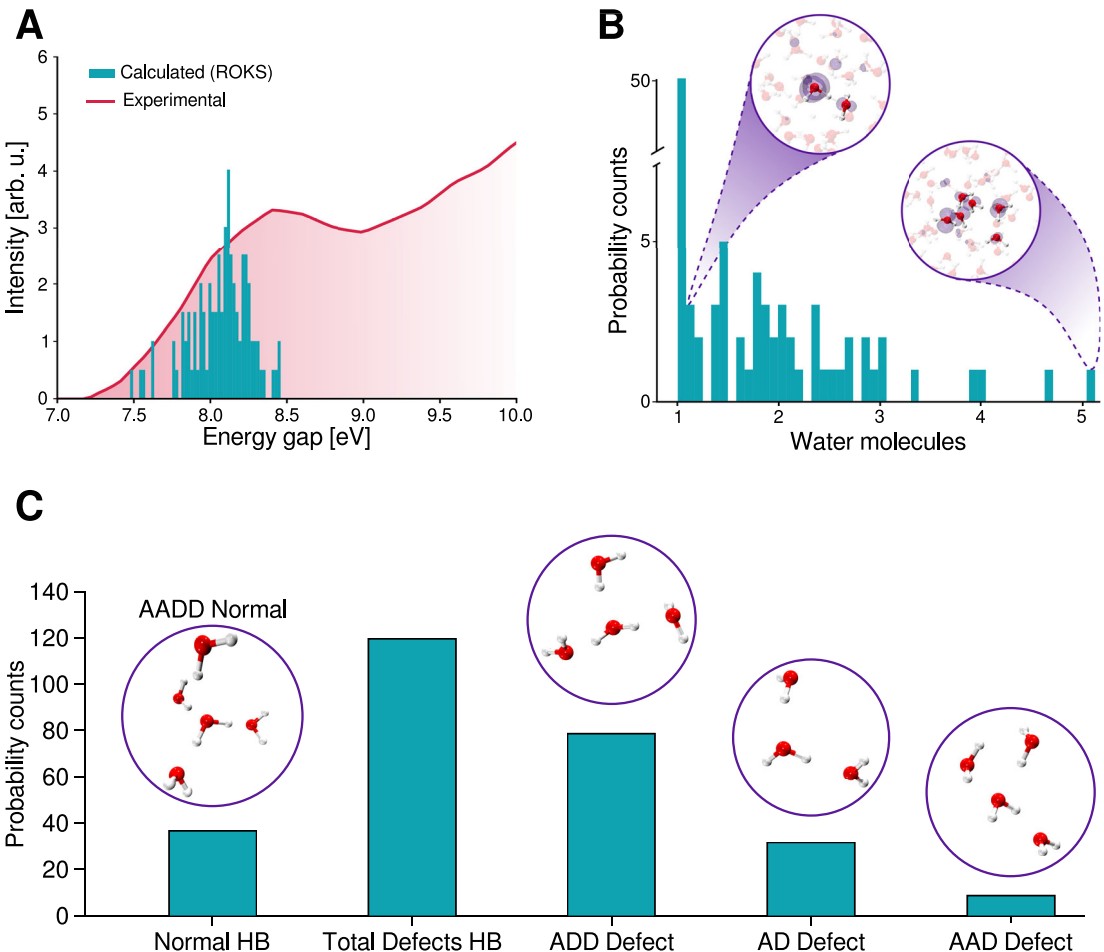

**Fig. 1 | Photo-initial absorption of neat liquid water. A** Experimental absorption spectra in solid red line (reproduced from the ref. [18]) and the calculated $S_0 \rightarrow S_1$ excitation using ROKS for 100 different conformations obtained from the electronic ground state. **B** Probability Counts of water molecules involved in the excitation using the Inverse Participation Ratio of the spin densities for all the conformations. The insets show two examples involving 1 and 5 water molecules. **C** Probability Counts of the different defects in the Hydrogen-Bond (HB) Network of all the water molecules involved in the initial excitation. The insets show the molecular geometries of the normal and each type of defect in the HB network. Source data are provided as a Source Data file.

molecular orbitals, electronic densities, or spin densities represented as isosurfaces, which can be highly sensitive to the chosen isovalue and thus somewhat arbitrary[28]. Our approach overcomes these limitations providing a quantitative description of the excitation which allows also for fractional contributions of the number of water molecules. In addition, previous studies have argued that the first transition is localized on a single water, however most of them were limited to isolated monomers or small water clusters[29,31,32], where the system size inherently restricts delocalization. Other studies have employed periodic water systems of varying boxes sizes, inferring localization from the weak dependence of excitation energies on the system size[30]. While these approaches often produce excitation energies and oscillator strengths that align well with experimental results, they generally rely on limited sampling and thus fail to capture the influence of thermal fluctuations on the excitation process. Understanding the degree of electronic delocalization upon excitation to the first band is particularly important, as one and two photon absorption experiments can access different electronic configurations[17,23,25]. These configurations, in turn, may lead to distinct pathways in the excited state dynamics of liquid water.

A comprehensive understanding of the first electronic transition in liquid water requires not only the characterization of electronic density but also the analysis of the surrounding environment. In this context, the directionality of the hydrogen-bond (HB) network around

the water molecules involved in the transition is particularly important. The analysis shown in Fig. 1B enables precise identification of the water molecules participating in the excitation, allowing us to classify them according to their HB environment. Figure 1C presents the probability counts of HB defects across the ensemble, along with a schematic representation of typical and defective hydrogen bonding motifs (see "Methods" section for details).

Our results clearly show that the water molecules most frequently involved in the lowest electronic transition are those found in hydrogen-bond (HB) environments with defects, particularly those with at least one missing acceptor hydrogen-bond. These include configurations such as AD (a water molecule accepting and donating a single hydrogen-bond) and ADD (a water molecule donating two and accepting one hydrogen bond), as illustrated in Fig. 1C. This observation is rooted in the nature of the electronic transition, which corresponds to a transition from a non-bonding orbital ($n$) to an antibonding orbital ($\sigma^*$). In a well-formed HB network, the non-bonding orbitals are stabilized through their participation in hydrogen bonding. This stabilization raises the energy required for excitation[29,31]. In contrast, when a water defect does not accept a hydrogen-bond, the non-bonding orbitals are no longer stabilized by the field from the nearby proton and therefore remain at higher energy. As a result, the energy gap for the $n \rightarrow \sigma^*$ transition becomes smaller. These arguments are essentially also at the root of the blue shift of about 0.7 eV observed

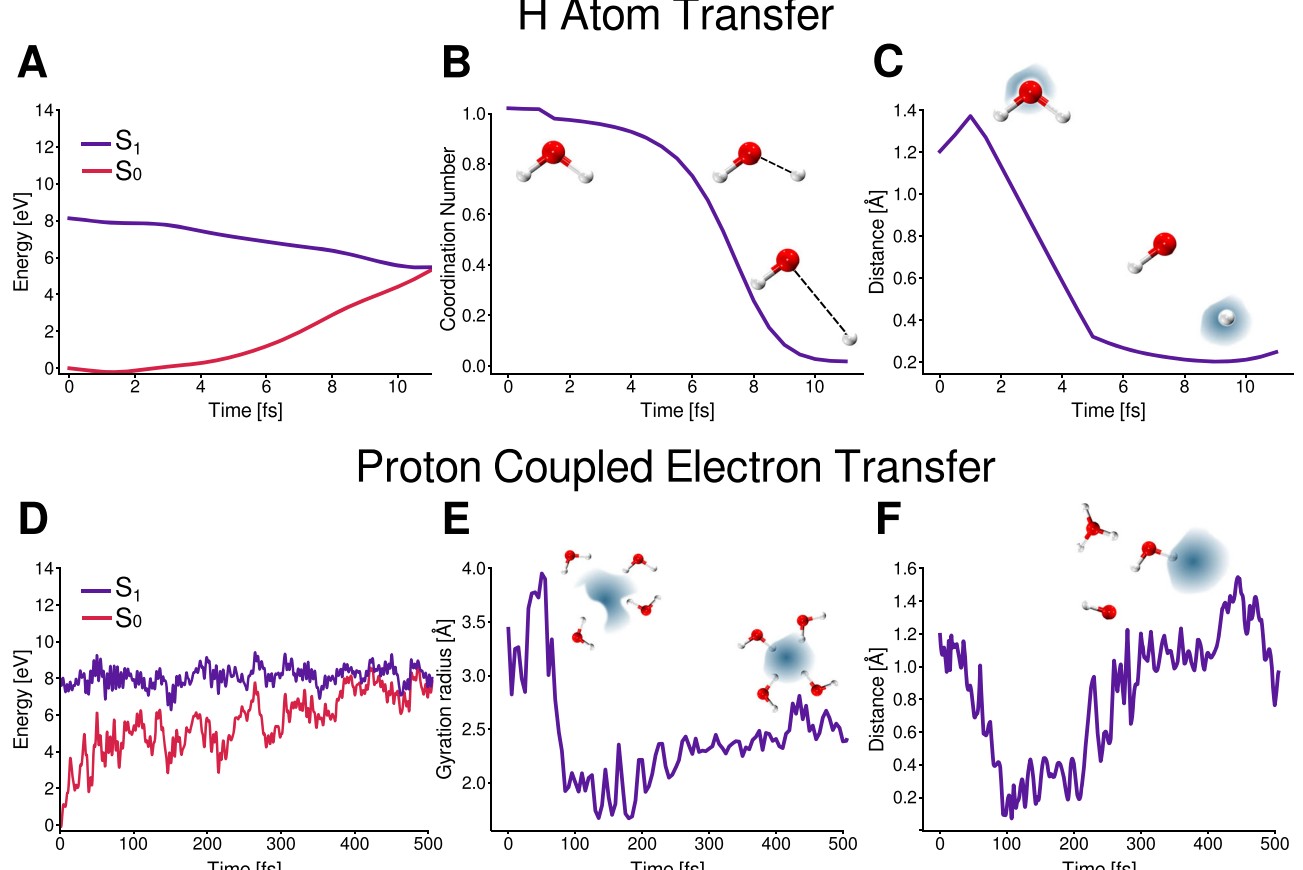

**Fig. 2 | Photo-physical decay mechanisms in neat liquid water upon excitation to the $S_1$ electronic state.** The upper panels show the H Atom Transfer mechanism and the lower panels the Proton Coupled Electron Transfer mechanism. **A**, **D** evolution of the potential energies of the $S_1$ and $S_0$ electronic states along a representative trajectory for both mechanisms. **B** shows the evolution of the coordination number of the hydrogen that is dissociating in the excited state dynamic. **C** evolution of the distance between the electron center and the closest H. **E** shows the evolution of the gyration radius of the electron. **F** shows the evolution of the distance between the electron center and the closest H. Source data are provided as a Source Data file.

in the first absorption band of water when comparing the liquid phase to the gas phase[18,19].

**Photo-chemical mechanism of water deactivation**

The excited state dynamics of liquid water reveal two distinct decay pathways: an ultrafast process with an average lifetime of ~25 fs, and a slower one of ~400 fs (see Supplementary Fig. 4). Our protocol, described in detail in the "Methods" section, defines the decay time as the instant when the energy gap between the $S_1$ and $S_0$ states drops below 0.2 eV. This approach has been successfully applied to describe excited state relaxation pathways in other systems[64,65] and, as we will show later, leads to insights that reinforce and enrich the interpretations of existing experimental observations. Before analyzing the full ensemble, we illustrate these mechanisms using representative trajectories in order to build our intuition on the underlying processes.

Figure 2 depicts the two decay pathways: Hydrogen Atom Transfer (HAT, upper panels) and Proton Coupled Electron Transfer (PCET, lower panels). The HAT mechanism exhibits an ultrafast decay. As shown in Fig. 2A, which presents the evolution of the $S_1$ and $S_0$ potential energy surfaces, the system reaches the non-radiative $S_1 \rightarrow S_0$ crossing at ~10 fs in this trajectory. During the initial few femtoseconds, one water molecule undergoes O-H bond dissociation. This is evident in Fig. 2B, which shows the coordination number of the dissociating hydrogen over time which goes from 1 to 0 (see "Methods" section). This behavior is consistent with the strongly dissociative character of the $S_1(n \rightarrow \sigma^*)$ potential energy surface and agrees with previous theoretical and experimental findings[18,19,66]. To determine whether the

dissociating species is a hydrogen atom (H·) or a proton (H⁺), we determined the center of the electron and its distance to the dissociating hydrogen (see "Methods" section for details). As shown in Fig. 2C, this distance rapidly drops from 1.2 to below 0.2 Å, clearly indicating that the observed process involves hydrogen atom dissociation.

The PCET mechanism displays a slower decay, on the order of hundreds of femtoseconds (see Fig. 2D). In fact, 2 out of 100 trajectories do not exhibit non-radiative decay to the $S_0$ state, remaining on the $S_1$ for the entire 1 ps of the simulation. Following photoexcitation of a water molecule, in addition to the OH dissociation, an electron is released into the surrounding hydrogen-bond network. Figure 2E shows the evolution of the gyration radius of the electron over time (see "Methods" section for details), which serves as a proxy for its spatial delocalization. Initially, the electron is highly delocalized, but around 100 fs becomes increasingly localized. This behavior is consistent with experimental observations[34,67], which report a transition from a very delocalized electron (gyration radius above 20 Å) to a more localized wet electron, and finally to the fully hydrated electron in the ground state (gyration radius 2.4 Å).

A key feature of this mechanism is that it also involves an ultrafast O-H bond dissociation, consistent with the dissociative character of the $S_1$ ($n \rightarrow \sigma^*$) potential energy surface. However, unlike the H atom pathway, here the dissociated species is a proton (e.g., H⁺ or $H_3O^+$). This is evident from Fig. 2F, which shows the distance between the center of the electron and the closest hydrogen. Initially, this distance decreases (within the first 100 fs) but in contrast to the HAT

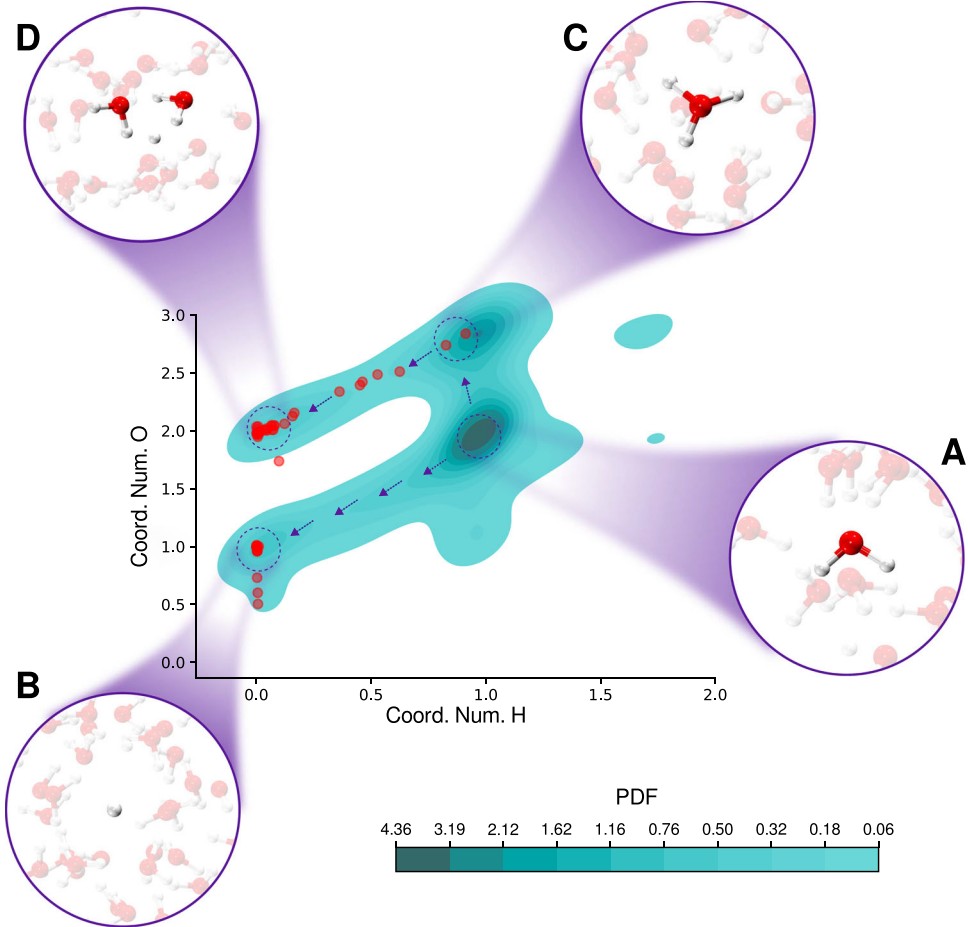

**Fig. 3 | H Atom Transfer Mechanism.** Two-dimensional probablity density function (PDF) of the coordination number of the closest H to the center of the electron (Coord. Num. H) and the coordination number of the closest O to this H (Coord. Num. O). Red circles show the values for the conformations at the crossing $S_1 \rightarrow S_0$ transition. **A** shows the water molecule at the initial steps. **B** shows the H atom in a empty cavity. **Panel C:** shows the formation of the radical $H_3O^\bullet$. **D** shows the water molecules solvating the H atom. Source data are provided as a Source Data file.

mechanism, the energy gap is still high (around 2 eV, see Fig. 2D). After this period of time, this distance subsequently starts to increase supporting the notion of the formation of the hydrated electron and the $H_3O^+$. This also implies that the dissociated water molecule that lost the original proton is now an $HO^\bullet$ species. Later we will delve into the role of the $H_3O^+$ and $HO^\bullet$ on the photophysical properties.

Using the features described in Fig. 2, specifically the gyration radius, distance between the electron center and the closest hydrogen, we can clearly distinguish between the two decay mechanisms. The detailed protocol for this classification is provided in the "Methods" section. While a comprehensive discussion of each mechanism across the full ensemble of trajectories is presented in the following two sections, it is worth noting that our results indicate that ~53% of trajectories follow the HAT pathway, while the remaining exhibit the PCET mechanism. Experimental studies report a 45% occurrence of the HAT pathway at 6.7 eV excitation and 70% at 8.4 eV[68,69]. Our result lie between these two values, in excellent agreement with the fact that our simulations probe the red edge of the first absorption band (8.1 eV).

## H atom transfer mechanism

To fully characterize the hydrogen atom transfer mechanism, we constructed a two-dimensional probability density function (PDF) using two key descriptors: the coordination number of the dissociating hydrogen atom ($CN_H$), which always carries the electron, and the coordination number of its nearest oxygen neighbor ($CN_O$). The

analysis (see "Methods" section for details), based on all trajectories that follow this mechanism, is presented in Fig. 3.

The most probable configuration, with $CN_H = 1$ and $CN_O = 2$ (Fig. 3A), corresponds to the initial state of the simulation, where the hydrogen atom is still part of a neutral water molecule. Upon photoexcitation, we observe a fast dissociation of this hydrogen atom, accompanied by a sharp drop in the $S_1 \rightarrow S_0$ energy gap from approximately 8 eV to below 0.2 eV, leading to a non-radiative transition (as indicated by all the red points in the Fig. 3). In some trajectories, the dissociating hydrogen atom moves into an empty cavity (Fig. 3B), where its nearest oxygen is part of a hydroxyl radical ($HO^\bullet$), resulting in $CN_O = 1$ and leading to a non-radiative crossing to the ground state.

Another frequently observed pathway is the formation of a species in which the dissociating hydrogen atom binds to a water molecule, forming the radical hydronium ($H_3O^\bullet$), characterized by $CN_H \sim 1$ and $CN_O \sim 3$ (Fig. 3C). We confirmed that this species is indeed $H_3O^\bullet$ and not the conventional $H_3O^+$ ion, since the electron remains associated with it. While the species $H_3O^\bullet$ has been predicted by various theoretical models[28,56,57,70], our work is the first to demonstrate its formation following excitation by simulating the entire process from photoabsorption to relaxation in liquid water. According to our simulations, the radical $H_3O^\bullet$ appears only transiently. It eventually undergoes further dissociation, where the hydrogen atom carrying the electron migrates into a region surrounded by water molecules. In these configurations, an OH bond from a nearby water molecule points

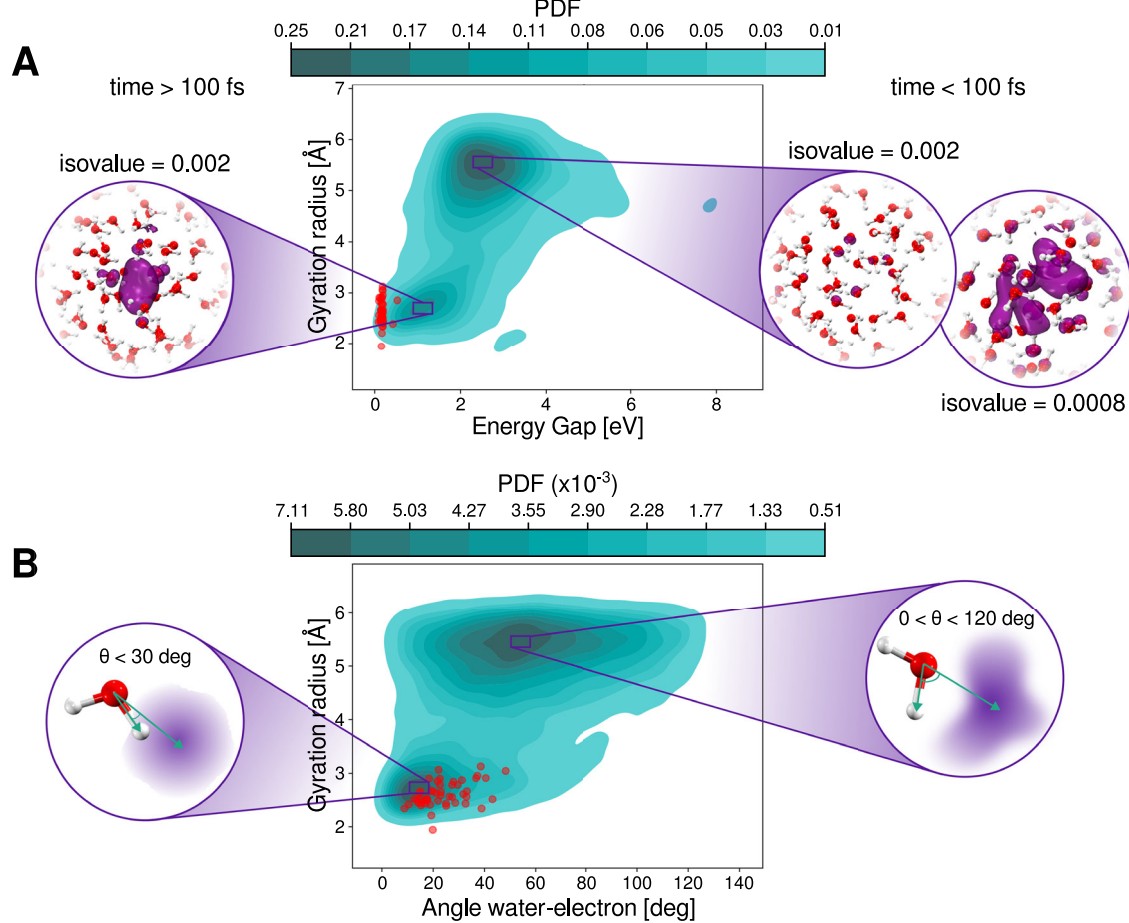

**Fig. 4 | Proton Coupled Electron Transfer Mechanism. A** Probability Density Function (PDF) of the gyration radius and the energy gap for all the trajectories. Red circles represent the conformations at the crossing point. Insets show the electronic spin density for representative conformations of the system at two different times during the excited state dynamics and at different iso-values. **B** Probability Density Function (PDF) of the gyration radius and the angle formed between the vector pointing from the Oxygen to the electron center and the vector of the O-H of the waters in the first solvation shell. Red circles represent the conformations at the crossing point. Insets show schematic representations of the water at the two gyration radius along with the vectors considered for the analysis. Source data are provided as a Source Data file.

toward the hydrogen atom (see Fig. 3D) creating an unstable geometry ultimately leading to non-radiative decay to the ground state. To the best of our knowledge the radical hydronium ($H_3O^\bullet$) has not been observed experimentally through photoionization/photoabsorption of water perhaps in part due to the fact that it is formed fleetingly as a rare transition state-like motif.

An interesting observation is that ~20% of the trajectories follow the mechanism involving the formation of a hydrogen atom in an empty cavity (Fig. 3B) with a lifetime of 13 fs, while the remaining 33% correspond to the alternative HAT pathway (Fig. 3C, D) exhibiting a longer lifetime of 34 fs. These results correlate well with the defect analysis presented in Fig. 1C. When the dissociating water molecule has a missing hydrogen-bond donor (AD or AAD defect), there is sufficient space available to allow an ultrafast hydrogen atom dissociation into a cavity (~ 13 fs), in excellent agreement with values reported for a hydrogen-bonded water dimer in vacuum[71,72]. In contrast, when the dissociating molecule lacks a hydrogen-bond acceptor (ADD defect), the surrounding hydrogen-bond network in the condensed phase restricts this motion, delaying the non-radiative decay and promoting the formation of the transient $H_3O^\bullet$ species.

### Proton Coupled Electron Transfer Mechanism
Our results indicate that this pathway leads to the formation of the hydrated electron while the system is still in the excited state. Figure 4

shows data compiled from all the frames and all the trajectories that follow this mechanism after photoexcitation of liquid water. Panel 4**A** presents a two-dimensional probability density plot of the gyration radius of the electron versus the energy gap between the $S_1$ and $S_0$ states. As described previously for a representative trajectory, the electron is initially highly delocalized following excitation, with an average gyration radius of ~6 Å. As time progresses, the gyration radius decreases and stabilizes around 2.7 Å, indicating localization (see Supplementary Fig. 8).

Experimental studies have reported initial gyration radius of around 20–40 Å[34,67] immediately after photoabsorption. Although our method captures the same qualitative behavior, direct comparison with experiment is limited by the size of the simulation box (12.42 Å), which restricts the maximum observable extent of delocalization[73]. The final gyration radius we observe in the excited state is 2.7 Å, slightly larger than the experimentally reported value of 2.4 Å for the hydrated electron in the ground state[74]. Note that the decrease of the gyration radius from 6 to 2.4 Å incurs a decrease in the energy gap. Once the electron has localized, the system undergoes a non-radiative transition to the ground state, as indicated by the red points in Fig. 4A.

As mentioned in the introduction, generating a free electron in bulk water using photon energies within the first absorption band requires some degree of nuclear rearrangement to release the electron from the molecule[17]. Earlier, we showed that this release occurs

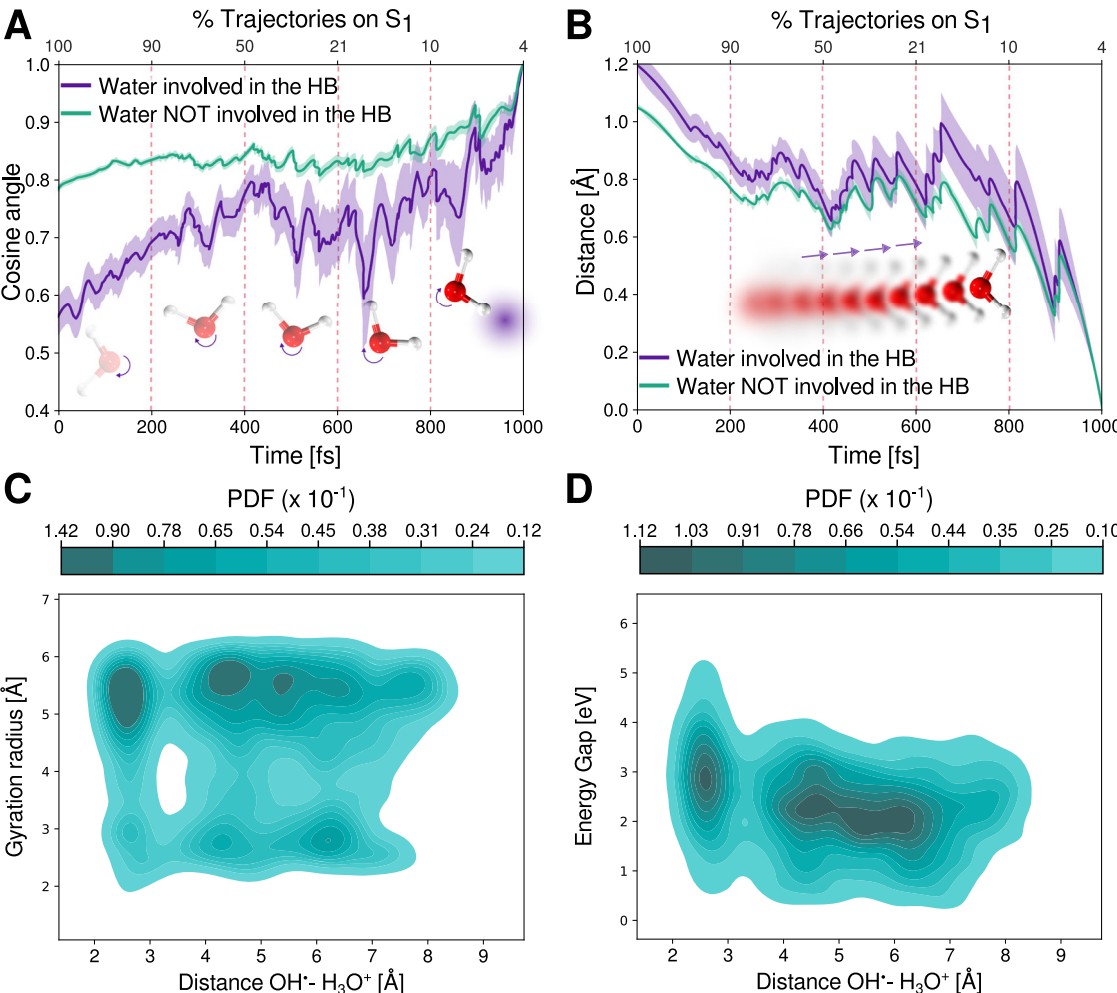

**Fig. 5 | Collective motions in the excited states dynamics. A** Average cosine angle between the dipole vector of each water with respect to the same vector at the end of the simulation. The waters that end up solvating the electron are plotted in purple and the rest of waters in green. Shadows represent the standard error (SE) across all the trajectories and the waters. **B** Average diffusion distance of each water molecule. The waters that end up solvating the electron are plotted in purple and the rest of waters in green. Shadows represent the standard error (SE) across all the trajectories and the waters. The upper axes represents the percentage of trajectories that remain in $S_1$ between all the trajectories undergoing the PCET mechanism. **C** 2D density plot between the gyration radius and the distance between the radical HO$^{\cdot}$ and the ion H$_3$O$^+$. **D** 2D density plot between the Energy gap $S_1$ and $S_0$ and the distance between the radical HO$^{\cdot}$ and the ion H$_3$O$^+$. The SE was calculated using $SE = \frac{\sigma}{\sqrt{N}}$ with the data coming from an analysis of 100 trajectories. The error bars reported represent those coming from trajectories that remain in the $S_1$ state. Source data are provided as a Source Data file.

simultaneously with the dissociation of the OH bond in the excited water molecule, leading to the formation of a free electron, a hydroxyl radical (HO$^{\cdot}$), and a proton (H$^+$or H$_3$O$^+$). We now move on to elucidating that the localization of the electron and the formation of the other species require a collective process involving reorganization of the hydrogen-bond network of the water molecules.

In Fig. 4B we illustrate the coupling between the gyration radius against the angle formed between the vector connecting the oxygen and the center of the electron and the OH vector of the water molecule. For this analysis we only include the nearby water molecules involved in the first solvation shell of the electron, (see "Methods" section for details). When the electron is delocalized, this angle spans a broad range from 0 to 120°, reflecting the absence of a well-defined solvation shell. As the system relaxes, the angle narrows to below 30°, consistent with the orientational preferences seen in hydrated electron solvation shells[58,73,75]. These results indicate that the structure of the hydrated electron forms within the excited state itself, before the non-radiative decay consistent with recent experiments using sub-two-cycle visible-to-shortwave infrared pump-probe spectroscopy[76]. Analysis of the radial distribution function in the low gyration radius regime on the

excited state (see Supplementary Fig. 9), where the hydrated electron is effectively localized, reveals that the first solvation shell comprises between 4-5 water molecules. This is consistent with previous findings for the hydrated electron in the ground electronic state[25,45,51].

## Coupled translational and rotational solvent dynamics stabilizes water-mediated photoproducts

The previous analysis does not directly capture the solvent dynamics that drive the transition toward a localized hydrated electron. To better understand the molecular rearrangements leading to the $S_1 \rightarrow S_0$ crossing configuration, we analyzed both the rotational and translational motion of water molecules during the excited state evolution.

Figure 5 A shows the time evolution of the cosine angle between each dipole vector of the water molecules and its dipole orientation at the end of the simulation, where the hydrated electron is fully localized. This measure quantifies how much each molecule reorients relative to its final configuration. We distinguish between water molecules that ultimately form the first solvation shell of the electron and those that remain farther away. For the population of waters that are eventually hydrogen bonded to the excess electron, we observe

**A**

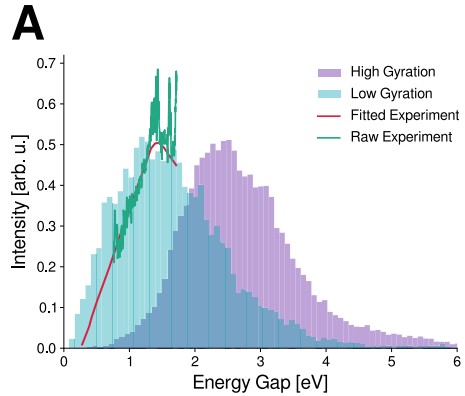

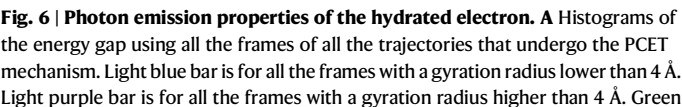

**B**

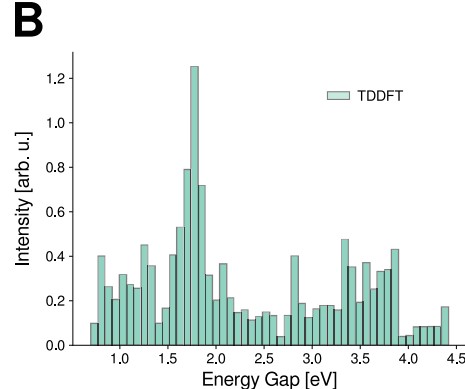

**Fig. 6 | Photon emission properties of the hydrated electron. A** Histograms of the energy gap using all the frames of all the trajectories that undergo the PCET mechanism. Light blue bar is for all the frames with a gyration radius lower than 4 Å. Light purple bar is for all the frames with a gyration radius higher than 4 Å. Green and red solid lines are the experimental and fitted fluorescence spectra of the hydrated electron. Experimental data were digitalized from ref. 58. **B** Fluorescence spectra using the oscillator strength and energy gap for the last 300 fs for this trajectory using TDDFT. Source data are provided as a Source Data file.

two phases of the dynamics. In the first part that occurs within the first 400 fs the dipole rotates by ~ 20 degrees, this leads to a reduction of the radius of gyration from 5 to 4 Å. This initial timescale fundamentally corresponds also to the creation of the initial cavity that will host the excess electron. On a timescale beyond 500 fs, further rotation of these water molecules leads to the final localization of the excess electron.

In addition to these rotational dynamics, Fig. 5B shows that localization also involves collective translational motion. Water molecules in the first solvation shell move on average by ~1.2 Å relative to their final positions, a displacement slightly larger than that of the more distant water molecules. Overall, our results show that both rotational and translational rearrangements of specific water molecules are necessary for the localization of the hydrated electron. These collective motions also cause a notable distortion of the hydrogen-bond network, consistent with experimental findings from ionization studies in liquid water[35]. A similar reorganizational response has been reported in ab initio molecular dynamics simulations of carbon dioxide reduction in water[77,78], where the breaking of hydrogen bonds in the second solvation shell triggers the stabilization of the excess electron on the carbon dioxide molecule. This process also involves coupled translational and rotational motions of water molecules, closely paralleling the behavior observed in our simulations.

As alluded to earlier, the creation of the excess electron after photoexcitation also leads to the formation of a hydronium ion ($H_3O^+$) and a hydroxyl radical (HO·). To investigate the behavior of these species during the excited state dynamics, we present in Fig. 5C the coupling between the gyration radius of the electron and the distance between the HO· and the $H_3O^+$. Interestingly, our analysis reveals three distinct minima in the HO·($H_3O^+$) distance, particularly in the region of low gyration radius where the electron becomes localized. These minima occur at ~ 2.6, 4.3, and 6.2 Å. Our simulations are in excellent agreement with very recent time-resolved electron diffraction experiments following the ionization of liquid water, where the corresponding minima were found at 2.4, 3.4, and 5.8 Å[35]. These three minima correspond to situations involving the initial formation of contact ion-radical pair and then the separation mediated by the solvent. Although previous theoretical studies have identified the HO·($H_3O^+$) pair by simulating ionized water with the electron removed[79–81], our findings show for the first time, the creation of these correlated structures upon photoexcitation starting from the neutral bulk water.

The creation of ion-radical pairs at different distances has important implications on the optical properties. To probe this, we show in Fig. 5D the energy gap as a function of the HO·($H_3O^+$) distance. For the contact ion-radical pair at 2.6 Å the average energy gap is ~ 3 eV on average. As one moves towards the creation of the solvent-separated ion-radical pairs, the energy gap drops to ~ 2.0 eV. This trend is also consistent with experimental observations, which demonstrated that the formation and spatial separation of the HO·($H_3O^+$) pair occur before the internal conversion associated with the $p \to s$ transition of the hydrated electron[35].

**Spontaneous emission from hydrated excess electrons**

An important and still not fully understood property of the hydrated electron in liquid water is its photon emission following excitation. Understanding this emission can provide valuable insight into the nature of the excited state and the relaxation dynamics of the system. Although experimental studies are relatively scarce, Tauber and Mathies have explored this optical behavior through fluorescence and Raman spectroscopies[58]. Motivated by their findings, we now focus on analyzing the emission spectrum obtained from our simulations and examine how it depends on the gyration radius of the hydrated electron.

Figure 6 A shows the experimental emission spectrum along with our calculated $S_1 \to S_0$ energy gaps, evaluated for configurations with both large and small electron gyration radius. As the excess electron becomes more localized, transitioning from a larger to a smaller gyration radius, the energy gap decreases from an average of 2.4 to 1.3 eV (516 to 954 nm). Interestingly, the experimental emission spectrum is in excellent agreement with our predictions in both peak position and spectral width, especially for configurations with smaller gyration radius. This observation supports previous interpretations from experimental[58] and theoretical studies[82,83], suggesting that emission occurs from a broad ensemble of transient excited state geometries rather than from a single well-defined minimum. It is important to highlight that our theoretical spectrum reflects the recombination of the electron, since we describe the final electronic ground state as a closed shell (see "Methods"). In contrast, the experimental measurements capture emission from the excited state of the hydrated electron without identifying the final state[58], which could involve either recombination or hydrated electron in the ground state. Therefore, while the agreement between our results and the experimental data is encouraging, it should be considered preliminary. A more detailed investigation of these optical properties is necessary and lies beyond the scope of this study. However, our findings provide a valuable framework that could guide future experimental and theoretical efforts aimed at confirming and better understanding the emission behavior of the hydrated electron.

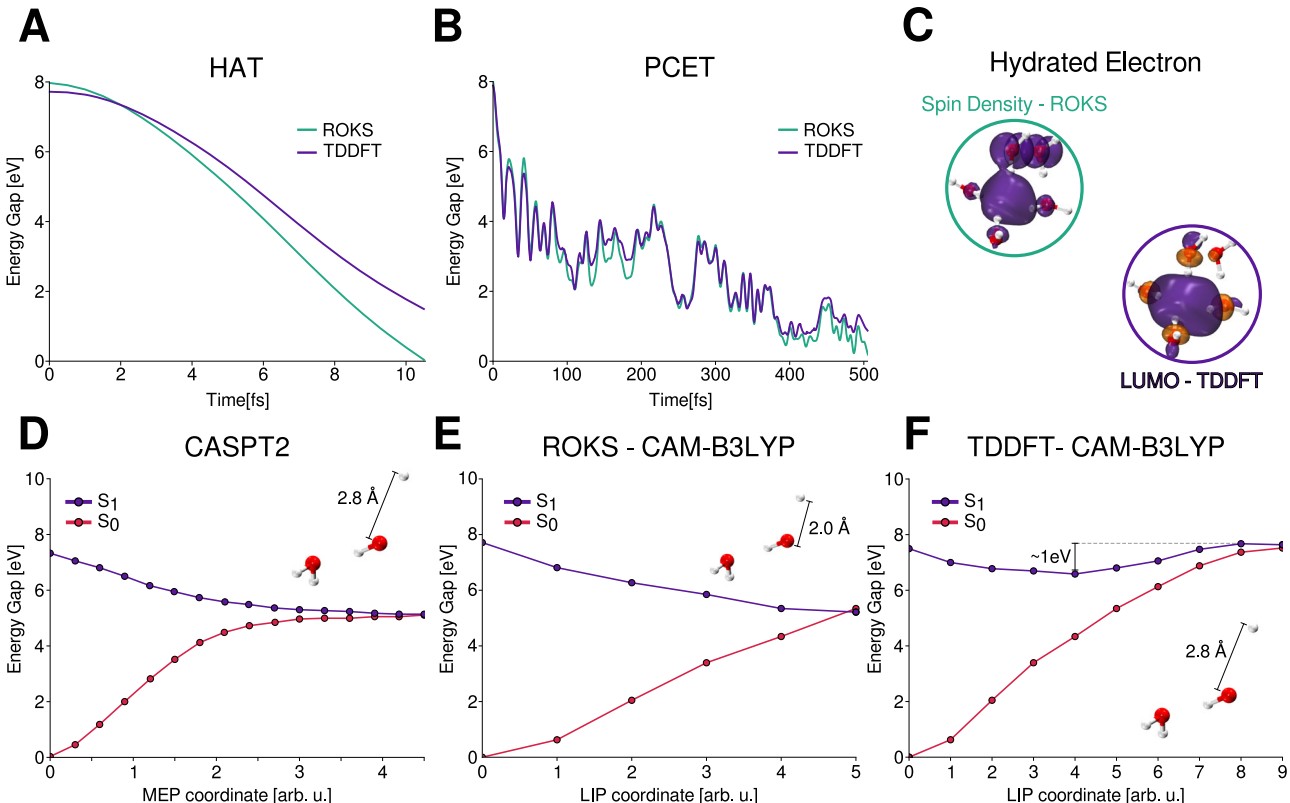

**Fig. 7 | Reliability of the ROKS approach for describing the photochemistry of water. A** Evolution of the energy gap for a representative HAT trajectory obtained with ROKS (green) and TDDFT (purple). **B** Evolution of the energy gap for a representative PCET trajectory obtained with ROKS (green) and TDDFT (purple). **Panel C:** Spin density from ROKS and LUMO orbital from TDDFT at 250 fs of the PCET trajectory, illustrating the localization of the hydrated electron. **D** Minimum Energy Path (MEP) for a water dimer in vacuum exhibiting the HAT mechanism, obtained with CASPT2. The inset shows the geometry at the conical intersection between $S_1$ (purple) and $S_0$ (red). Data reproduced from ref. 71. **Panel E:** Linear Interpolated Path (LIP) for the same water dimer in vacuum exhibiting the HAT mechanism, obtained with ROKS-CAM-B3LYP. The inset shows the geometry at the $S_1/S_0$ conical intersection. **Panel F:** Linear Interpolated Path (LIP) for the water dimer exhibiting the HAT mechanism, obtained with TDDFT-CAM-B3LYP. The inset shows the geometry at the $S_1/S_0$ conical intersection. Source data are provided as a Source Data file.

In order to ensure that these features are not sensitive to the choice of our ROKS method used for the simulation of the excited state, we computed the emission spectrum for the last 300 fs of a representative PCET trajectory using the energy gaps and oscillator strengths obtained from TDDFT (Fig. 6B). While the spectrum is not fully converged due to limited sampling, the resulting peak at 1.7 eV closely matches ROKS calculated and experimental emission peak. Additional validations for both mechanisms are provided in the next section.

## Evaluating the accuracy of ROKS for the photochemistry of liquid water

In this work, we employed the ROKS approach to investigate the photochemistry of liquid water. While our results show excellent qualitative and quantitative agreement with a broad range of experimental observations, it is important to assess the quality of the electronic structure method and to evaluate its performance relative to more extensively tested theoretical approaches in photochemistry.

In order to first assess the robustness of ROKS with respect to the choice of exchange-correlation functional and basis set, we performed additional calculations using the CAM-B3LYP functional[84] and the TZV2P basis set[85]. The results, presented in Supplementary Note 9, demonstrate that the ROKS method provides consistent energy gaps and reproduces the same behavior for both photochemical pathways (HAT and PCET), confirming its reliability across different electronic structure settings.

We further compared our results with the TDDFT method, which has been widely applied to the study of excited states and in non-adiabatic dynamics. In addition, for a hydrogen-bonded water dimer in vacuum, we benchmarked our approach against both multi-configurational methods and TDDFT. The main results are summarized in Fig. 7.

Panel 7A,B show the evolution of the energy gap between the first excited state ($S_1$) and the ground state ($S_0$) for two representative trajectories (sampled from the ROKS dynamics), corresponding to the HAT and PCET mechanisms, obtained using both ROKS and TDDFT with the same functional and basis set employed in this work (see Methods section). At the initial step, both methods yield nearly identical energy gaps, as further confirmed by a broader validation across all initial conditions (see Supplementary Fig. 1).

The main difference arises in the HAT mechanism. At the end of the simulation, ROKS predicts a crossing point with an energy gap smaller than 0.2 eV, whereas TDDFT yields a value near 1.5 eV for the same geometry. Despite this quantitative difference, both methods describe the same physical process: water dissociation leading to the formation of HO· and H·. In contrast, the PCET mechanism shows excellent agreement between ROKS and TDDFT. In this case, both methods predict the dissociation of a water molecule into HO· and $H_3O^+$ within the first 20 fs, with comparable energy gap profiles. Moreover, TDDFT also reproduces the localization of the hydrated electron, consistent with the ROKS results, as shown in Fig. 7C, which compares the ROKS spin density and TDDFT LUMO at 250 fs.

To further understand the discrepancy between ROKS and TDDFT observed for the HAT mechanism, we examined a model system consisting of a hydrogen-bonded water dimer in vacuum[71,72]. Segarra et al. computed the Minimum Energy Path (MEP) for the photodissociation of one water molecule from the Franck-Condon region to the $S_1/S_0$ conical intersection using the multiconfigurational CASPT2 method (see Fig. 7D). Using both the ground state and conical intersection optimized geometries at this high level of theory, we computed the potential energy surfaces along the Linear Interpolated Path (LIP) with ROKS (Fig. 7E) and TDDFT (Fig. 7F) employing the CAM-B3LYP functional. It is worth emphasizing that the numerical values along horizontal axes for the MEP and LIP coordinates do not correspond to identical nuclear configurations, except for the initial point shown in Fig. 7D–F. We thus focus on the overall trends of the ground and excited state potential energy surfaces.

ROKS predicts a dissociative pathway along the $S_1$ surface, similar in slope to CASPT2. The main difference lies in the position of the crossing point: at a HO·H· distance of ~2 Å for ROKS, compared with 2.8 Å for CASPT2. Although an O-H distance of 2 Å could be considered as dissociated geometry, the topology of the conical intersection differs between the two methods. Despite this difference, ROKS reproduces the same essential physical process, namely the hydrogen atom dissociation. In contrast, TDDFT predicts a local minimum on the $S_1$ surface at around 2 Å with an energy gap of ~1.5 eV, which is consistent with the differences seen in the condensed phase simulations for the HAT mechanism (Fig. 7A). Moreover, TDDFT exhibits a ~1 eV barrier from this minimum to the conical intersection, which could significantly delay, or even prevent, reaching the intersection within the timescales (~10 fs) reported in previous studies[71,72].

Taken all together, these results provide a comprehensive validation of the ROKS approach. We demonstrate that ROKS captures the essential physics of both the HAT and PCET photochemical pathways in liquid water. Based on the gas phase water dimer model, ROKS does not reproduce the conical intersection geometry for the HAT mechanism. In particular, a difference of approximately of 0.8 Å is observed in the HO·H· distance at the conical intersection when compared with CASPT2. Nevertheless, ROKS qualitatively describes the same photodissociation pathway, reproducing both the slope of the excited state potential energy surface and the presence of the relevant chemical species along the dissociation coordinate. While such differences may lead to deviations in the predicted HAT lifetimes in vacuum, we expect their impact to be reduced in the condensed phase. Indeed, in bulk liquid water, ROKS yields lifetimes and quantum yields that are in good agreement with experimental observations. Overall, the extensive validation presented here demonstrates the robustness of our methodology and reinforces confidence in its predictive capability for modeling the excited state dynamics and photochemistry of liquid water. We nonetheless caution that more systematic studies with multireference methods for condensed phase aqueous systems needs to be explored in the future.

## Discussion

In this work, we have investigated the complex photochemistry of liquid water following excitation to its first absorption band. Our simulations fill a critical knowledge gap in modeling the initial photoexcitation event and the pathways that lead to the creation of the excited state hydrated electron. Specifically, we began by quantifying the degree of electronic delocalization upon photoexcitation and uncovered how structural defects and fluctuations in the hydrogen-bond network modulate transition energies, particularly in the low energy region of the band. By tracking the excited state dynamics, we identified all reactive species observed experimentally, such as the hydrated electron, H atom, hydroxyl radical, hydronium ion and/or radicals, and finally, motifs involving water-mediated ion-radical pairs. Our results highlight the crucial role of ultrafast collective rearrangements of water molecules in controlling both the evolution and localization of the hydrated electron, as well as the fate of other transient species.

We find that most excitations to the first band are localized on a single water molecule, but a significant fraction are delocalized over up to five molecules. This behavior, intrinsically linked to the asymmetry of the hydrogen-bond network, may explain the distinct photochemical outcomes observed at different energies within the first absorption band[18,68,69]. Our excited state molecular dynamics simulations reveal two distinct photophysical decay pathways: Hydrogen Atom Transfer (HAT) and Proton Coupled Electron Transfer (PCET). For the HAT mechanism, we predicted lifetimes and quantum yields that are in excellent agreement with experimental data. Moreover, we observed the formation of the hydronium radical, a transient species previously proposed in theoretical studies but, to the best of our knowledge, not yet observed experimentally and reported here for the first time starting from photoexcited pure liquid water. This species has been suggested as a possible precursor to the hydrated electron[18,76]. While our simulations, performed in the electronic excited state, cannot exclude the possibility that it could release an electron in the ground electronic state, further studies are needed to confirm this pathway.

In the PCET pathway, the hydrated electron emerges in the excited state. Our results further reveal that this localization is driven by ultrafast (<1 ps) collective rotational and translational motions of water molecules, particularly those participating in hydrogen bonding with the electron. In its early stages, the hydrated electron is highly delocalized. However, as localization proceeds, the gyration radius approaches values consistent with experimental measurements for the equilibrated hydrated electron in the electronic ground state. These findings indicate that a significant degree of electron localization together with collective waters rearrangements, occurs in the excited state before non-radiative decay. Moreover, we also observe the formation of hydroxyl radical-hydronium ion pairs, which was recently observed experimentally[35]. We investigated how their spatial separation influences both the gyration radius of the hydrated electron and the evolving energy gap during the excited state dynamics. This correlation is particularly relevant in the early stages of the process, when diffusion is limited and the interactions between species are strongest.

Finally, our work provides new insights into the broad fluorescence emission associated with the hydrated electron, as reported in experiments. We demonstrate a clear correlation between the emission energy and the degree of localization of the electron. This relationship offers a fresh perspective on the optical behavior of the hydrated electron. In recent studies[86–88], fluorescence emission has been observed in different water solutions. Although different mechanisms have been proposed, the hydrated electron has not been considered as a contributing factor underlying these experimental observations.

In conclusion, our theoretical studies here have provided fresh insights into the underlying mechanisms associated with one the most fundamental processes in the chemical physics and physical chemistry of water, namely how light interacts with liquid water and the subsequent photochemistry. Through a synergy of data-driven approaches, our approach lays the ground-work to study the birth of the excited state hydrated electron in different bulk aqueous salt solutions[86–88], at aqueous interfaces[14,89] and various biological contexts such as radiation-induced DNA damage[4,5]. On the more methodological side, it is interesting to explore whether our simulations can be used to develop machine-learning approaches[90–94] to examine the excited state dynamics of aqueous systems. Such methods would enable simulations with larger unit cells and a greater number of trajectories, potentially improving agreement with experimental observations and revealing new physical insights.

## Methods

### Ground state molecular dynamics

We began by constructing a periodic simulation box containing 64 water molecules with a side length of 12.42 Å, corresponding to a density of 1 $g/cm^3$ at room temperature. An initial equilibration was performed using an empirical water model, SPC/E[95] with the GROMACS software[96] within the NVT ensemble for 100 ns. The canonical-sampling velocity-rescaling[97] thermostat was applied with a coupling time of 0.1 ps.

From this classical equilibrated trajectory, we selected 100 configurations, separated by 1 ns, as initial conditions to perform ab initio quality ground state molecular dynamics simulations with a machine-learning interatomic potential (MLIP). Specifically, we chose an MLIP recently trained and developed using the DeePMD-kit architecture[98] on data computed using Density Functional Theory (DFT) with the SCAN0 functional[99], which incorporates 10% exact exchange to mitigate the self-interaction error. Prior studies[99] have shown that SCAN0 provides an excellent description of the structural, dynamical, and electronic properties (such as IR spectra, electronic band structure, and band gap) of bulk liquid water compared to the original SCAN meta-GGA functional[100,101]. Each of the 100 simulations conducted with the MLIP potential was run for 1 ns to generate well-converged and decorrelated initial conditions (positions and velocities), leveraging DeePMD-kit[98] in combination with the LAMMPS package[102].

All in all, our ground state sampling protocol ensures that we have statistically uncorrelated initial conditions (nuclear positions and velocities) for the water hydrogen-bond network, which is essential for accurately describing the excited state dynamics.

### Excited state molecular dynamics

The excited state molecular dynamics (ESMD) simulations were initiated from the first electronic excited state ($S_1$), computed using the Restricted Open-shell Kohn-Sham (ROKS) method[46,47,103]. This method provides an alternative approach for calculating singlet excited states in closed-shell systems that are described by non-Aufbau electronic configurations. By minimizing the energy expression $E_S = 2E_{mix} − E_T$, where $E_S$, $E_T$, $E_{mix}$ correspond to the energies of the singlet, triplet, and mixed states, respectively, the method effectively accounts for a two-determinant nature of the excited state. Consequently, ROKS offers a computationally efficient and reliable framework for describing dissociative and charge-transfer excited states[48–50,104].

To propagate the system on the $S_1$ potential energy surface, we employed the Atomic Simulation Environment (ASE)[105], a python module which is interfaced with CP2K software[106]. At each timestep, we calculated the energy on the electronic ground state $S_0$ using Density Functional Theory (DFT) to account for possible non-radiative relaxation pathways[49,51] (see below) which in our current setup is determined by examining the energy gap. The simulations were performed in the ensemble NVE using a timestep of 0.5 fs and velocity verlet algorithm to perform the nuclei propagation of the system.

Both excited and ground state calculations were performed using the hybrid PBEh(40)-rVV10 functional[107] in combination with the DZVP-MOLOPT-SR basis set[85]. This level of electronic structure theory has been successfully applied to the study of liquid water[108–110], demonstrating its accuracy in reproducing ionization potential, redox potentials and electronic levels of excess electron. Excellent agreement with the experiments for aqueous iodine solutions were reported by Carter et al.[51] within ROKS framework and by Lan et al.[111] employing a similar approach with the same functional.

More recently the functional PBEh(40) was identified as one of the most reliable choices for describing the hydrated electron in its electronic ground state[73]. In our current work, the favorable agreement with reproducing the experimental UV-absorption and other quantities sensitive to the excited states of liquid water gives us confidence in the quality of our results.

Each ESMD simulation was propagated until the system reached a crossing point, defined as the step where the energy gap between the first excited state ($S_1$) and the ground state ($S_0$) dropped below 0.2 eV. This criterion provides a robust and efficient way to identify non-radiative decay events and allows us to characterize the distribution of deactivation pathways, offering a lower bound on the excited state lifetimes[64,65]. A known limitation of this approach is that it does not explicitly include non-adiabatic coupling vectors (NACVs), which are key quantities for describing electronic transitions during non-radiative decay. However, a previous study on the radiolysis of water cluster in vacuum showed that NACVs increase significantly as the energy gap decreases[112]. This strong correlation implies that small energy gaps are reliable indicators of high non-adiabatic transition probabilities. Therefore, by using the energy gap as a decay criterion, we effectively capture the relevant non-radiative dynamics, justifying the omission of explicitly determining NACVs in our simulations[113,114]. To further support this approximation, we evaluated the Landau-Zener (LZ) transition probabilities[115], which in addition to the energy gap criterion, also account for the curvature of the potential energy surfaces (see Supplementary Note 10). This approach has been shown to reproduce the results consistent with those obtained using explicit NACVs in non-adiabatic dynamics[115–119].

The simulations that did not exhibit a decay were continued until a maximum time of 1 ps was reached. Spin density cube files were calculated and outputted every 5 fs during all ESMD trajectories and at the step of crossing, for post-processing analysis (see section below). All the electronic calculations in the present work were performed using CP2K version 2024.2[106].

### Absorption and emission spectra

The absorption spectrum to the first band and emission spectrum presented in this work (Figs. 1A and 6A, respectively) were obtained by calculating the energy gap between $S_1$ and $S_0$ using the methods described in the previous section. Because the CP2K software does not provide the transition dipole moments within our approach, the spectra were represented as histograms of the calculated energy gaps. This approach was validated through the calculation of energy gap and oscillator strength using TDDFT with the same functional and basis set, as shown in Supplementary Note 1 for the absorption spectrum and in Fig. 6 for the emission spectrum.

### Coordination number analysis

To quantify local structure and bond-breaking/formation occurring within the hydrogen-bond network, we determined the coordination numbers of hydrogen ($CN_H$) including only the oxygen atoms in close vicinity and the coordination number of the oxygen ($CN_O$) including only hydrogens in close vicinity. We employed the following smooth switching function:

$$CN = \sum_{i>0} \mathbf{S}(|\mathbf{r}_i − \mathbf{r}_0|) \quad \mathbf{S}(\mathbf{r}) = \frac{1}{exp[\kappa(\mathbf{r} − \mathbf{r}_c)]+1} \qquad (1)$$

Here, $i$ is the atom index, while $\kappa = 10$, $\mathbf{r}_c = 1.38$ Å are parameters taken from previous works used to study the dissociation of water in the ground state to form hydronium and hydroxide ions[120–123]. In this manner, for an isolated hydrogen atom not attached to any water molecule, the $CN_H$ is zero while for a hydrogen covalently bonded to a water molecule $CN_H$ is 1. In the case of $CN_O$, it takes on a value of 3, 2 and 1, for a hydronium (ion or radical), neutral molecular water and hydroxyl species respectively.

### Hydrogen-bond (HB) network classification

HB were identified using a geometrical criteria introduced by Kumar et al.[124]: O-O distance of ≤3.5 Å and H-O−O angle ≤30$deg$. A water molecule with four HB is classified as part of a normal HB network,

otherwise, it is labeled as a defect (see Fig. 1C for categories). Sensitivity analysis to the choice of these geometric parameters is addressed in Supplementary Note 3 showing that while the hydrogen-bond criteria naturally affect the exact populations of defects, the trends and physical insights inferred are robust.

## Spin-density analysis for excitation localization

To identify which water molecule was initially excited, we analyzed the electronic spin density for all the 100 initial configuration. We decomposed the total spin density by integrating around each oxygen atom within a radius of 1 Å.

$$s_i = \int_{|r-r_0^i| \leq R} \rho(r)dr \tag{2}$$

where $r_0^i$ is the position of the $i-th$ oxygen atom and $R = 1$ Å. To estimate the degree of localization, we computed the Inverse Participation Ratio (IPR)[52,53]:

$$IPR^{-1} = \left( \sum_i s_i^2 \right)^{-1} \tag{3}$$

This metric provides a continous measure of how many water molecules participate in the excitation. The distribution of $IPR^{-1}$ values for all the initial configurations is shown in Fig. 1B. The sensitivity analysis and validation to the choice of the integration radius parameter are discussed in Supplementary Note 2.

## HAT versus PCET Mechanism

Photo-excitation of liquid water follows either one of these two mechanisms:

$$H_2O \xrightarrow{h\nu} HO^\bullet + H^\bullet \text{ (\textbf{HAT})} \tag{4}$$

$$H_2O \xrightarrow{h\nu} HO^\bullet + H^+ + e^- \text{ (\textbf{PCET})} \tag{5}$$

A direct calculation of the electron center from the spin density cube files yields a position located between the HO˙ and the H˙ (Equation (4)), or between the HO˙ and the excess electron (Equation (5)). Supplementary Fig. 5 illustrates this point. This result lacks a clear physical interpretation, as it does not correspond to the actual location of any species and essentially implies that the spin density is located close to the geometric center of the species involved. This problem does not arise in earlier theoretical studies[25,28,45,75], as those simulations typically start with a pre-inserted electron in bulk water and the formation of an additional radical species are not observed.

To overcome this challenge, we first identify the HO˙ species by locating the water molecule with the longest OH bond in each trajectory (see Supplementary Note 6). Once identified, we remove the contribution of the HO˙ to the spin density by excluding a spherical region of radius 1.5 Å centered on its oxygen atom (see Supplementary Note 7). This step yields a modified cube file that provides a more meaningful representation of the spin density to localize the hydrated excess electron. We then compute the electron center and gyration radius from this file using the protocol described in previous studies[45] (see Supplementary Note 8). Since the cube files were saved every 5 fs and at the crossing geometry, the electron center and gyration radius for the intermediate frames were obtained using linear interpolation between consecutive cube files.

Once all the species were correctly identified, we classified each trajectory according to its reaction mechanism. For this, we analyzed the final frame of each simulation (corresponding to the geometry at the point of non-radiative crossing) and we computed the distance between the electron center and the nearest hydrogen atom. If the distance was less than 0.6 Å, the trajectory was classified as hydrogen atom transfer (HAT), otherwise it was assigned to the proton coupled electron transfer (PCET). The 0.6 Å threshold was chosen based on the radial distribution function (RDF) between the electron and all hydrogen atoms (see Supplementary Fig. 9). Specifically, it represents the midpoint between two cases: a bound hydrogen atom (distance electron and nearest H ~ 0 Å) and the first maximum associated with hydrogen atoms in the solvation shell of the electron (distance between electron and nearest H ~ 1.2 Å)[45,51]. A short electron-hydrogen distance indicates that both species occupy the same spatial region, consistent with the formation of a neutral hydrogen atom. In contrast, larger distances suggest that the electron remains solvated while the proton is bound to nearby water molecules, consistent with a PCET process.

Based on the preceding analysis, each trajectory can be unambiguously assigned to a specific mechanism. This allows us to determine the quantum yield and lifetime associated with each mechanism using the following expression:

$$QY_M = \frac{\# \text{ Trajectories with M}}{\#\text{Total trajectories}} * 100 \tag{6}$$

$$\tau_M = \frac{\sum_i^{N_M} t_i^M}{\# \text{Total trajectories with M}} \tag{7}$$

where $M$ represent the mechanism (HAT or PCET), $QY$, $\tau$ are the quantum yields and lifetimes, $N_M$ is the number of trajectories with the $M$ mechanism and $t_i^M$ is the time at which the trajectory $i$ with the $M$ mechanism finds the "crossing point".

## Data availability

All data generated in this study, including ESMD trajectories, cube files for representative pathways, geometries for the water dimer, input files for all calculations, and scripts for the analysis of spin density, have been deposited in the Zenodo database (https://doi.org/10.5281/zenodo.17714130). Source data are provided with this paper.

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

## Acknowledgements

G.D.M., D.D and A.H. acknowledge funding from the European Research Council (ERC) under the European Union's Horizon 2020 research and innovation program (grant agreement No. 101043272 - HyBOP). The views and opinions expressed are those of the authors only and do not necessarily reflect those of the European Union or the European Research Council Executive Agency. Neither the European Union nor the granting authority can be held responsible for them. G.D.M., C.M. and A.H. also acknowledge MareNostrum5 (project EHPC-EXT-2023E01-029) for computational resources. C.J.M. was supported by the U.S. Department of Energy (DOE), Office of Science, Basic Energy Sciences (BES), the Chemical Sciences, Geosciences, and Biosciences Division, Chemical Physics and Interfacial Sciences Program, FWP 16249.

## Author contributions

G.D.M. and A.H. conceived and designed the research. C.M. performed the ground state MLIP simulations, and G.D.M. carried out the ESMD simulations. G.D.M., S.D.P., C.K.E., and D.D. performed the data analysis. G.D.M., C.J.M., and A.H. contributed to the interpretation and discussion of the results. G.D.M. and A.H. wrote the original manuscript with input from all authors. All authors discussed the findings and contributed to the final version of the manuscript.

## Competing interests

The authors declare no competing interests.
