## [Transparent Peer Review file · Nature Communications]

Simulating the Photochemical Birth of the Hydrated Electron in Liquid Water

Corresponding Author: Dr Gonzalo Díaz Mirón

Version 0:

Reviewer comments:

Reviewer #1

(Remarks to the Author)

In the manuscript *Ultrafast Solvent Dynamics Drives the Formation of the Hydrated Electron in Photoexcited Water*, the authors reported a study on the excited dynamics of photoexcited water, which is important in understanding the formation of the hydrated electron. By running excited-state molecular dynamics, the manuscript discovered two possible pathways after the liquid water becomes photoexcited: hydrogen atom transfer (HAT), which does not form the hydrated electron in the excited state, and proton-coupled electron transfer (PCET), which undergoes localization of the hydrated electron. The authors performed a detailed analysis of their results and provided solid evidence supporting their findings. I recommend it to be published after minor revisions, which may address my following questions or concerns:

1. The S1 excited state is simulated using the ROKS scheme, as in the authors' previous CTTS study [J. Phys. Chem. Lett. 2023, 14 (4), 870–878]. Similar CTTS simulations have also been reported in [Nat. Commun. 15, 2544 (2024)]. The S1 state could alternatively be simulated using LR-TDDFT. Have the authors attempted to perform the dynamics using LR-TDDFT for comparison?
2. In line 104 – 105, the authors stated that their prediction that the HAT mechanism exhibits a higher quantum yield is consistent with experimental observations. Also, in line 106 – 107, the authors mentioned that the hydronium radical has not been experimentally confirmed. Then how is the quantum yield of the HAT mechanism determined?
3. In line 203, the authors stated that 2 out of 100 trajectories remain in the S1 excited state. It seems confusing to me. As I understand, the system always stays on the S1 excited state, and one tracks the energy gap between S1 and S0. The system can be the S1 to S0 crossing in most trajectories, but this work does not consider the hopping from S1 to S0. If this is correct, every trajectory should remain in the S1 state. Do the authors mean that the 2 out of 100 trajectories do not reach S1 to S0 crossing here?
4. In Section 2.3, the authors showed different geometries of the HAT products, i.e., Figures 3B and 3D. Are they both observed as the final state at the S1 to S0 crossing in the MD trajectories? If so, what is the ratio between these two products? Can the authors use their simulation results to discuss a bit on which of these two geometries is favored? Moreover, in the HAT pathway, will the hydrated electron form after the non-radiative decay to the ground state?
5. In line 254 – 257, the authors mentioned that the unstable geometry, as shown in Figure 3D, ultimately leads to non-radiative decay to the ground state. Does this imply that the non-radiative decay is only associated to the unstable geometry in Figure 3D? Does the geometry where the hydrogen atom enters an empty cavity, i.e., Figure 3B, also undergo the non-radiative decay? From Figure 2A, it seems that cavity geometries might also decay, but the authors connected the decay with the unstable geometry, which is confusing.
6. For consistency, the caption of Figure 4 might start with “Proton Coupled Electron Transfer Mechanism”. The “Translation” in the title of Section 2.5 might be changed to “Translational”

Reviewer #2

(Remarks to the Author)

The manuscript "Ultrafast Solvent Dynamics Drives the Formation of the Hydrated Electron in Photoexcited Water" by G. Díaz Mirón et al is devoted to simulating excited state dynamics of liquid water in order to reveal initial stages of solvated electron's formation. By ab initio molecular dynamics and careful data analysis the authors characterize two main processes after photoexcitations: hydrogen atom abstraction (HAB) and proton-coupled electron transfer. HAB involves H3O radical as a transient species, never characterized before, which comprises considerable novelty. PCET is found to lead to the formation of the solvated electron and is driven by the solvated dynamics. The research is carefully designed, thoroughly executed and clearly reported. It provides considerable novelty already at the stage of asking a question - how is solvated electron formed by light? - which has not been addressed by previous studies. The answer appears to be convincing, too. At the same time I have several serious methodological concerns, which need to be addressed:

1) My gut feeling and personal experience tell that early stages of the solvated electron's formation has not been considered by theoreticians not out of curiosity shortage, but rather due to the lack of proper methodologies. ROKS with hybrid DFT used here is a cheap and efficient solution, which undeservedly remained in shadows. It exhibits excellent performance for excitation energies of liquid water as shown here. Nevertheless, its ability to describe bond breaking is far less obvious. On the contrary, spin-restricted theories (namely ROHF, a parent method of ROKS) is notorious for pathological behavior for bond-breaking. Instead, one uses spin-unrestricted variants of SCF, poor-man's strong correlation for chemical reactions. ROKS is not exactly ROHF, but I have never seen any unequivocal evidence for its performance for dissociation. The authors state that this is demonstrated in the references, however, I could find that only ref 50 has to do with hydrogen dissociation, and it does not look conclusive as benchmarking ROKS was not the focus. Therefore, it would be important to see at least energy profiles of hydrogen abstraction from a water molecule or a water dimer to be able to assess the performance of ROKS.

2) "Later" stages of the trajectories considered here show that S0 and S1 states become close in energy for both processes, these stretches of dynamics being relatively long for PCET. It implies that non-adiabatic effects must play considerable role here. I won't suggest to remake everything with surface-hopping approaches, but it's an important issue. Actually computation of non-adiabatic couplings in CP2K is now possible due to the work of A. Hehn et al.

3) Spin densities in ROKS are not clearly defined: CP2K writes .cube files but issues a warning in the output. And indeed, the total multiplicity is singlet and strictly speaking there is no spin density despite the presence of unpaired electrons (AKA singlet diradicals). Please, clarify the point.

A minor point:

It may be instructive to compare the role of solvent dynamics in solvated electron's formation with the role of solvent dynamics in its reaction with CO2. It appears that there may be useful analogies and generalization. See:

<https://pubs.acs.org/doi/abs/10.1021/acs.jpcc.0c07859>

and

<https://pubs.acs.org/doi/10.1021/acs.jpcc.3c06935>

Reviewer #3

(Remarks to the Author)

The ms. by Miron et al. deals with ultrafast dynamics following photoexcitation in liquid water, specifically focusing on the formation of the hydrated electron. Using Restricted Open-Shell Kohn Sham (ROKS) molecular dynamics simulations, the authors aim to understand the initial steps following photoexcitation in water. They explore in detail the initial excitation localization and the subsequent hydrogen atom transfer and proton-coupled electron transfer pathways, and how this is related to formation and relaxation of the hydrated electron. In this way, the authors paint a detailed picture of how photoexcitation of water initiates a cascade of events, ultimately leading to the formation of the hydrated electron through a specific interplay of proton and electron transfer, coupled with the dynamic restructuring of the hydrogen-bond network. Moreover, they manage to successfully link their results from their DFT dynamics to experimental observations, wherever available.

This is a well-executed and insightful study that provides a valuable contribution to our understanding of the photodynamics of liquid water, that should be interesting to a broad readership. Therefore, the ms. should in my view be published after a minor revision that reflects the issues raised below.

1. The accuracy of the ROKS results depends heavily on the functional used, basis set, and how well the method captures dynamic correlation. A more thorough assessment of the performance for excited-state dynamics in photoionized water, would be valuable.

2. I would appreciate a more detailed description how exactly the ROKS method is used for excited state dynamics. Comparison to other methods such as TDDFT would be valuable.

3. The authors acknowledge the simplification of not explicitly including non-adiabatic couplings. While it may be reasonable for the present system to consider the energy gaps only, a more rigorous justification or exploration of the potential impact of the approximation used on the results would strengthen the argument.

4. Given the computational demands of the method, it is understandable that the authors used a relatively small unit cell with limited sampling of initial configurations. Nevertheless, at least some discussion on the size and sampling effects would be appropriate.

5. I am puzzled by the author's finding of the H3O radical. This must be a very unstable species (and the authors indeed

discuss its transient character) but I wonder if it is not more like a $\text{H}_3\text{O}^+\dots\text{e}^-$ contact "ion pair".

Reviewer #4

(Remarks to the Author)

The authors describe a simulation study for studying the formation of the hydrated electron. This process is a benchmark chemical process and has been studied experimentally and theoretically by many groups. Here the authors present a study which follows the pathway from the photoexcitation into the first absorption band all the way.

They describe two distinct pathways: Hydrogen Atom Transfer (HAT) and Proton Coupled Electron Transfer (PCET). The predictions for the HAT mechanism are in excellent agreement with the experimental time constants.

Their data analysis involves quantification of the hydrogen bond network defect, the localization of the solvated electron. Investigations of the H Atom Transfer and predictions of the photo emission properties of the hydrated electron.

I have the following comments:

Their key analysis tool is to use "dilution factor" or "inverse participation ratio" to quantify how many water molecules an excess electron is delocalized over.

Note: strictly speaking the "PDFs" in Fig. 1BC are counts, not probability distribution functions; they should be normalized, or maybe called probability counts.

The conclusion is almost complete localization: the peak in Fig. 1B for 1 water is 50, at 2 water molecules we are already down to ~2 counts, or 4%. This needs to be mentioned more clearly. So, the further analysis does not change the view compared to with one water. The figures in Fig. 2B could be clearer, they are hard to see on the screen.

Defects account for $\frac{3}{4}$ of the binding sites, mainly missing acceptor sites: a missing acceptor bond means the water molecule has a lone pair on oxygen not bonded to an H from another water molecule, destabilizing the n to σ^* transition to ~8 eV. What is missing here is a further aspect: the steric room available next to an unsatisfied acceptor bond also helps the $\text{HO} + \text{H}$ dissociation process by making room for the recoiling OH and H.

The definition of the decay lifetime on p. 9 is somewhat arbitrary but consistent, sufficient to distinguish processes differing by more than a factor of 10 in lifetime. Maybe this definition can be made more precise.

In the PCET mechanism ($n \rightarrow \sigma^* \text{H}_2\text{O} + \text{H}_2\text{O} \rightarrow \text{HO} + \text{e}^- + \text{H}^+ + \text{H}_2\text{O} \rightarrow \text{HO} + \text{e}^- + \text{H}_3\text{O}^+$), the e^- quickly becomes localized in a trapped state. The OH and H_3O^+ become quite well separated (6 Å in Fig. 5D).

Their theoretical study reveals three distinct minima in the HO^* and (H_3O^+) distance which are in nice agreement with the study of the recent electron diffraction experiments upon photoionization. These could be rationalized by the formation of contact ion radical pair and the separation mediated by the solvent.

The results are not surprising, the new aspect is that the entire process can be modelled from photoexcitation up to the localized electron directly capturing the solvent dynamics.

Reviewer #5

(Remarks to the Author)

Version 1:

Reviewer comments:

Reviewer #1

(Remarks to the Author)

I thank the authors for thoroughly addressing my comments. I recommend publication of the manuscript in Nature Communications.

In addition, it would be helpful if the authors could clarify the following point:

In Figures 1B and 1C, the y-axis is labeled "Probability counts", which is not normalized and therefore does not sum to 1. However, the caption states that these panels show the probability distribution function, which implies a normalized quantity. The figure would be easier to interpret if this inconsistency were resolved.

Reviewer #2

(Remarks to the Author)

In the revised version of the manuscript "The Photochemical Birth of the Hydrated Electron in Liquid Water" the authors have addressed main issues pointed out in the first round of peer review: provided validation of ROKS for the HAT mechanism, discussed and evaluated the importance non-adiabatic effects and performed simple non-adiabatic (Landau-Zener theory). In addition, they referred to more relevant publications.

I will support the publication of this work in Nature Communications if the discussion of the benchmark of ROKS against CASPT2 is done properly and fairly: their evaluation of the methods is way too optimistic, contradicting the data. Unfortunately, I have reasons to believe it is not the case. Panels D and E in Figure 7 show what I was afraid such a benchmark would show: very significant qualitative differences. Yes, the barrier height is indeed similar, but the geometry of TS is embarrassingly different (which is ingeniously, and reasonably, downplayed). However, the values in between the reactant and the conical intersection are nowhere near. Say at the coordinate value of 2 (units are not given at the axis, please add it) the the CASPT2 energy is ca. 2 times larger than that of ROKS, the difference being immense 2 eV. The shape of conical intersection region is dramatically different: it is clear from the figure that non-adiabatic effects at the multireference level of theory will be significantly larger since the gap squared appears in the exponential in the Landau-Zener theory.

I believe it's non proper to write about "close agreement with TDDFT and CASPT2" and so on in the last paragraph of Section 3. Instead, the authors should come up with realistic evaluation of their approach as at most semiquantitative.

Reviewer #3

(Remarks to the Author)

The authors satisfactorily addressed my comments, so I believe the ms. may be published.

Reviewer #4

(Remarks to the Author)

This is a theoretical simulation study on the birth of a solvated electron following all the steps thorough from the photoexcitation ,i.e. from the ground state to the excited state and then the formation of the hydrated electron up to localization including radical formation.

New experimental techniques have been developed including e.g. ultrafast electron diffraction which allows to monitor this process experimentally. Thus a theoretical study which involves all steps is of relevance and has not been reported before.

The data analysis and the choice of the theoretical methods should be checked in detail by theoretical referees.

Being outside of this field I can only state that the results are interesting and noteworthy.

Reviewer #5

(Remarks to the Author)

Comments for: Ultrafast Solvent Dynamics Drives the Formation of the Hydrated Electron in Photoexcited Water

November 2025

Dear Reviewers,

We sincerely appreciate the time and effort that the Reviewers have dedicated to evaluating our manuscript and providing thoughtful and constructive feedback. One of the main concerns raised pertains to the validation of our computational approach specifically the use of ROKS to study the excited-state dynamics of liquid water. While the original version of the manuscript already included a validation against TD-DFT results, we fully agree with the editors and reviewers that this aspect required a much more extensive and detailed treatment.

Before providing detailed responses to each of the reviewers' comments, we would like to briefly summarize the main changes that have been implemented in the revised manuscript. This has involved conducting significant benchmarking and validation with new calculations including:

- An assessment of the robustness of the ROKS approach using different basis sets and exchange-correlation functionals.
- Validation of ROKS against TDDFT for both the HAT and PCET mechanisms.
- To gain deeper insights, we additionally benchmark ROKS and TDDFT results against multiconfigurational CASPT2 calculations for a hydrogen-bonded water dimer in vacuum.
- Finally, we also determine the Landau–Zener (LZ) transition probabilities, which incorporate not only the energy gap but also the curvature of the potential energy surfaces.

For the convenience of all Reviewers, we have reproduced below the new section added to the revised manuscript (pages 23-26), which incorporates most of the modifications discussed above.

3 Evaluating the Accuracy of ROKS for the Photochemistry of Liquid Water

In this work, we employed the ROKS approach to investigate the photochemistry of liquid water. While our results show excellent qualitative and quantitative agreement with a broad range of experimental observations, it is important to assess the quality of the electronic structure method and to evaluate its performance relative to more extensively tested theoretical approaches in photochemistry.

In order to first assess the robustness of ROKS with respect to the choice of exchange–correlation functional and basis set, we performed additional calculations using the CAM-B3LYP functional[1] and the TZV2P basis set[2]. The results, presented in section S9 in the Supplementary Information, demonstrate that the ROKS method provides consistent energy gaps and reproduces the same behavior for both photochemical pathways (HAT and PCET), confirming its reliability across different electronic structure settings.

We further compared our results with the TDDFT method, which has been widely applied to the study of excited states and in non-adiabatic dynamics. In addition, for a hydrogen-bonded water dimer in vacuum, we benchmarked our approach against both multiconfigurational methods and TDDFT. The main results are summarized in Figure 7.

Figure 7: *Reliability of the ROKS approach for describing the photochemistry of water. **Panel A:** Evolution of the energy gap for a representative HAT trajectory obtained with ROKS (green) and TDDFT (purple). **Panel B:** Evolution of the energy gap for a representative PCET trajectory obtained with ROKS (green) and TDDFT (purple). **Panel C:** Spin density from ROKS and LUMO orbital from TDDFT at 250 fs of the PCET trajectory, illustrating the localization of the hydrated electron. **Panel D:** Minimum Energy Path (MEP) for a water dimer in vacuum exhibiting the HAT mechanism, obtained with CASPT2. The inset shows the geometry at the conical intersection between S_1 (purple) and S_0 (red). Data reproduced from ref. 71. **Panel E:** Linear Interpolated Path (LIP) for the same water dimer in vacuum exhibiting the HAT mechanism, obtained with ROKS-CAM-B3LYP. The inset shows the geometry at the S_1/S_0 conical intersection. **Panel F:** Linear Interpolated Path (LIP) for the water dimer exhibiting the HAT mechanism, obtained with TDDFT-CAM-B3LYP. The inset shows the geometry at the S_1/S_0 conical intersection.*

Panels **7A** and **7B** show the evolution of the energy gap between the first excited state (S_1) and the ground state (S_0) for two representative trajectories (sampled from the ROKS dynamics), corresponding to the HAT and PCET mechanisms, obtained using both ROKS and TDDFT with the same functional and basis set employed in this work (see Computational Methods section). At the initial step, both methods yield nearly identical energy gaps, as further confirmed by a broader validation across all initial conditions (see Figure S1 in the Supplementary Information).

The main difference arises in the HAT mechanism. At the end of the simulation, ROKS predicts a crossing point with an energy gap smaller than 0.2 eV, whereas TDDFT yields a value near 1.5 eV for the same geometry. Despite this quantitative difference, both methods describe the same physical process: water dissociation leading to the formation of HO^\bullet and H^\bullet . In contrast, the PCET mechanism shows excellent agreement between ROKS and TDDFT. In this case, both methods predict the dissociation of a water molecule into HO^\bullet and H_3O^+ within the first 20 fs, with comparable energy gap profiles. Moreover, TDDFT also reproduces the localization of the hydrated electron, consistent with the ROKS results, as shown in Figure **7C**, which compares the ROKS spin density and TDDFT LUMO at 250 fs.

To further understand the discrepancy between ROKS and TDDFT observed for the HAT mechanism, we examined a model system consisting of a hydrogen-bonded water dimer in vacuum [3, 4]. Segarra et al. computed the Minimum Energy Path (MEP) for the photodissociation of one water molecule from the Franck–Condon region to the S_1/S_0 conical intersection using the multiconfigurational CASPT2 method (see Figure **7D**). Using both the ground-state and conical intersection optimized geometries at this high level of theory, we computed the potential energy surfaces along the Linear Interpolated Path (LIP) with ROKS (Figure **7E**) and TDDFT (Figure **7F**) employing the CAM-B3LYP functional.

ROKS predicts a dissociative pathway along the S_1 surface, similar in slope to CASPT2. The main difference lies in the position of the crossing point: at a $\text{HO}^\bullet\text{-H}^\bullet$ distance of approximately 2 Å for ROKS, compared with 2.8 Å for CASPT2. Although this difference is not insignificant, a 2 Å separation already indicates a fully dissociated configuration, confirming that ROKS and CASPT2 capture the same underlying photodissociation physics. In contrast, TDDFT predicts a local minimum on the S_1 surface at around 2 Å with an energy gap of ~ 1.5 eV, which is consistent with the differences seen in the condensed phase simulations for the HAT mechanism (Figure 7A). Moreover, TDDFT exhibits a ~ 1 eV barrier from this minimum to the conical intersection, which could significantly delay, or even prevent, reaching the intersection within the timescales (~ 10 fs) reported in previous studies^{71,72}.

Taken all together, these results provide a comprehensive validation of the ROKS approach. We demonstrate that ROKS captures the essential physics of both the HAT and PCET photochemical pathways in water, in close agreement with TDDFT and CASPT2 benchmarks. This consistency across different mechanisms and details of the underlying electronic structure supports the robustness of our methodology and reinforces confidence in its predictive capability for modeling the excited-state dynamics of liquid water.

We believe that these substantial additions significantly enhance both the rigor and clarity of the manuscript, further strengthening the reliability and impact of the results presented in this work. We also would like to clarify that we have decided to change the title of the manuscript to one more accurately reflects the scope and findings of the present study. The revised title is: *The Photochemical Birth of the Hydrated Electron in Liquid Water*. In the following sections, we provide detailed responses to each point raised by the Reviewers. The corresponding revisions are indicated in the file “manuscript_marked”, where page and line numbers are referenced for clarity.

Responses to Reviewer #1:

1.1: The S1 excited state is simulated using the ROKS scheme, as in the authors’ previous CTTS study [J. Phys. Chem. Lett. 2023, 14 (4), 870–878]. Similar CTTS simulations have also been reported in [Nat. Commun. 15, 2544 (2024)]. The S1 state could alternatively be simulated using LR-TDDFT. Have the authors attempted to perform the dynamics using LR-TDDFT for comparison?

We thank the Reviewer for this valuable comment. We fully agree that the S_1 state could, in principle, also be simulated using LR-TDDFT. Our decision to employ the ROKS approach was motivated by two main factors. Firstly, for the most part, until only very recently, LR-TDDFT for excited-state dynamics has been primarily used for clusters in the gas phase or embedded within a QM/MM setup. LR-TDDFT calculations in fully periodic systems have certainly been deployed to compute absorption spectra[5]. Thus the use of LR-TDDFT for excited-state dynamics in periodic systems is far from mainstream apart from some very recent exceptions[6]. Secondly, ROKS provides a computationally feasible alternative to LR-TDDFT and as evidenced in a previous CTTS study published in JPCL, it is a reasonable compromise. Our benchmarks and validation further show that ROKS is perfectly suitable for studying the excited-state dynamics of liquid water.

We also appreciate the Reviewer for bringing to our attention the recent Nature Communications paper, which we were unaware of at the time of the original submission. We have now cited this work in the revised manuscript, as it further supports and validates the methodology adopted in our study. The following modification has been made accordingly (page 30, lines 570-573):

Excellent agreement with the experiments for aqueous iodine solutions were reported by Carter et. al.⁵¹ within ROKS framework and by Lan et. al.¹¹¹ employing a similar approach with the same functional.

In the original version of our manuscript, we had already included a preliminary validation of ROKS against TDDFT, such as the absorption spectra computed for all initial conditions and the evolution of the energy gap along a representative PCET trajectory (see Figures S1 and 6B in the previous version, respectively). However, following the Reviewers’ suggestion for a more detailed validation, we have now substantially expanded this part of the study.

Specifically, we have added a new section titled *“Evaluating the Accuracy of ROKS for the Photochemistry of Liquid Water.”*(pages 23-26 in the revised manuscript). In this

section, we provide a comprehensive comparison between ROKS and LR-TDDFT. We include energy gap profiles for two representative trajectories (one for the HAT mechanism and one for the PCET mechanism) calculated using both methods. To further clarify the observed differences, we also performed benchmark calculations for a hydrogen-bonded water dimer in vacuum, comparing ROKS and TDDFT results with high-level multiconfigurational CASPT2 calculations. Since we show many explicit benchmarks for ROKS in the rest of the responses, the reviewer is encouraged to also examine the responses there.

These additional results provide a thorough validation of the ROKS approach and reinforce confidence in its ability to accurately describe the photochemistry of liquid water.

1.2: In line 104 – 105, the authors stated that their prediction that the HAT mechanism exhibits a higher quantum yield is consistent with experimental observations. Also, in line 106 – 107, the authors mentioned that the hydronium radical has not been experimentally confirmed. Then how is the quantum yield of the HAT mechanism determined?

We thank the Reviewer for raising this important point. We agree that the distinction between the experimentally detected species and those predicted in our simulations needs clarification.

The hydronium radical ($\text{H}_3\text{O}^\bullet$) is indeed part of our HAT mechanism. In this process, the excess electron remains localized near a hydrogen species, which can exist either as a free H atom (H^\bullet) or transiently as part of the hydronium radical ($\text{H}_3\text{O}^\bullet$). As explained in our manuscript, both cases correspond to the same mechanistic class, because the electron-hydrogen distance identifies an overall H-centered configuration (see *Analysis Protocols*). Experimentally, only the free H atom has been detected; however, our simulations suggest the formation of a short-lived intermediate, the $\text{H}_3\text{O}^\bullet$ radical, which to our knowledge has not yet been confirmed experimentally.

Regarding on how the quantum yield of the HAT mechanism is determined, we emphasize that in our computational framework the quantum yield is defined based on the number of trajectories that follow a given mechanistic pathway. Because each trajectory can be unambiguously assigned to either HAT or PCET, the quantum yield simply reflects the statistical occurrence of each mechanism within our ensemble. Additional information is provided for the different situations founded in HAT mechanism in our response to **point 1.4**, which closely relates to this question.

To make this clearer, we have revised the *Analysis Protocols* section to explicitly define the quantum yields and lifetimes (paged 35-36, lines 680-688) as follows:

... Based on the preceding analysis, each trajectory can be unambiguously assigned to a specific mechanism. This allows us to determine the quantum yield and lifetime associated with each mechanism using the following expression:

$$QY_M = \frac{\# \text{Trajectories with } M}{\# \text{Total trajectories}} * 100 \quad (6)$$

$$\tau_M = \frac{\sum_i^{N_M} t_i^M}{\# \text{Total trajectories with } M} \quad (7)$$

where M represents the mechanism (HAT or PCET), QY, τ are the quantum yields and lifetimes respectively, N_M is the number of trajectories with the M mechanism and t_i^M is the time at which the trajectory i with the M mechanism finds the "crossing point".

1.3: In line 203, the authors stated that 2 out of 100 trajectories remain in the S1 excited state. It seems confusing to me. As I understand, the system always stays on the S1 excited state, and one tracks the energy gap between S1 and S0. The system can be the S1 to S0 crossing in most trajectories, but this work does not consider the hopping from S1 to S0. If this is correct, every trajectory should remain in the S1 state. Do the authors mean that the 2 out of 100 trajectories do not reach S1 to S0 crossing here?

We thank the Reviewer for pointing out this ambiguity in our manuscript. The analysis of the Reviewer is correct, the statement "2 out of 100 trajectories remain in the S_1 excited state" refers to those trajectories in which the system does not reach the crossing condition within the 1 ps simulation window. In other words, these trajectories remain on the S_1 potential surface for the entire duration of the simulation.

To eliminate this confusion, we have revised the corresponding sentence in the manuscript (page 10, lines 205–207) as follows:

The PCET mechanism displays a slower decay, on the order of hundreds of femtoseconds (see Figure 2D). In fact, 2 out of 100 trajectories ~~remain in the S_1 excited state for the entire 1 ps simulation~~ do not exhibit non-radiative decay to the S_0 state, remaining on the S_1 for the entire 1 ps of the simulation.

1.4: In Section 2.3, the authors showed different geometries of the HAT

products, i.e., Figures 3B and 3D. Are they both observed as the final state at the S1 to S0 crossing in the MD trajectories? If so, what is the ratio between these two products? Can the authors use their simulation results to discuss a bit on which of these two geometries is favored? Moreover, in the HAT pathway, will the hydrated electron form after the non-radiative decay to the ground state?

We thank the Reviewer for raising this insightful question. As correctly pointed out, both conformations 3B and 3D (and in a few cases 3C) were indeed observed at the non-radiative $S_1 \rightarrow S_0$ crossing points in our simulations. In Figure 3, all red points represent the geometries corresponding to these crossing events. We have clarified this explicitly in the revised manuscript (page 13, line 246):

...leading to a non-radiative transition (*as indicated by all the red points in the Figure 3*)

Regarding the second part of the Reviewer’s question, we fully agree that a discussion comparing the different pathways within the HAT mechanism is valuable. Accordingly, we have expanded this section in the revised manuscript to include an analysis of the lifetimes and quantum yields associated with each photoproduct. The following changes were introduced in Section 2.3 (pages 14–15, lines 264-274):

... *An interesting observation is that approximately 20% of the trajectories follow the mechanism involving the formation of a hydrogen atom in an empty cavity (Figure 3B) with a lifetime of 13 fs, while the remaining 33% correspond to the alternative HAT pathway (Figures 3C and 3D) exhibiting a longer lifetime of 34 fs. These results correlate well with the defect analysis presented in Figure 1C. When the dissociating water molecule has a missing hydrogen-bond donor (AD or AAD defect), there is sufficient space available to allow an ultrafast hydrogen atom dissociation into a cavity (~ 13 fs), in excellent agreement with values reported for a hydrogen-bonded water dimer in vacuum^{71,72}. In contrast, when the dissociating molecule lacks a hydrogen-bond acceptor (ADD defect), the surrounding hydrogen-bond network in the condensed phase restricts this motion, delaying the non-radiative decay and promoting the formation of the transient $\text{H}_3\text{O}^\bullet$ species.*

Concerning the Reviewer’s final question, we acknowledge that for the HAT mechanism the hydrated electron could, in principle, form after the non-radiative decay to the ground state. However, our present simulations are restricted to the excited-state dynamics and do not explicitly include the possibility for the electron ejection process

to occur in the ground state. We already included this limitation in the Conclusions section (page 27, lines 488–490):

While our simulations, performed in the electronic excited state, can not exclude the possibility that it could release an electron in the ground electronic state, further studies are needed to confirm this pathway.

1.5: In line 254 – 257, the authors mentioned that the unstable geometry, as shown in Figure 3D, ultimately leads to non-radiative decay to the ground state. Does this imply that the non-radiative decay is only associated to the unstable geometry in Figure 3D? Does the geometry where the hydrogen atom enters an empty cavity, i.e., Figure 3B, also undergo the non-radiative decay? From Figure 2A, it seems that cavity geometries might also decay, but the authors connected the decay with the unstable geometry, which is confusing.

We thank the Reviewer for bringing this to our attention. We acknowledge that our original description may not have been sufficiently clear, which could have led to confusion. The Reviewer is indeed correct that both conformations 3B and 3D lead to non-radiative decay. We have revised the manuscript accordingly to clarify this point.

As mentioned in our response to **point 1.4**, we first clarified the meaning of all red points in Figure 3 (page 13, line 246).

...leading to a non-radiative transition (*as denoted by all the red points in the Figure 3*)

In addition, we explicitly state that conformation 3B also leads to non-radiative decay (page 13, lines 248–249).

... In some trajectories, the dissociating hydrogen atom moves into an empty cavity (Figure 3B), where its nearest oxygen is part of a hydroxyl radical (HO^\bullet), resulting in $CN_O = 1$ *and leading to a non-radiative crossing to the ground state.*

1.6: For consistency, the caption of Figure 4 might start with “Proton Coupled Electron Transfer Mechanism”. The “Translation” in the title of Section 2.5 might be changed to “Translational”

We thank the Reviewer for carefully pointing out these errors. Both issues have been corrected in the revised version of the manuscript.

Responses to Reviewer #2:

2.1: My gut feeling and personal experience tell that early stages of the solvated electron’s formation has not been considered by theoreticians not out of curiosity shortage, but rather due to the lack of proper methodologies. ROKS with hybrid DFT used here is a cheap and efficient solution, which undeserved remained in shadows. It exhibits excellent performance for excitation energies of liquid water as shown here. Nevertheless, its ability to describe bond breaking is far less obvious. On the contrary, spin-restricted theories (namely ROHF, a parent method of ROKS) is notorious for pathological behavior for bond-breaking. Instead, one uses spin-unrestricted variants of SCF, poor-man’s strong correlation for chemical reactions. ROKS is not exactly ROHF, but I have never seen any unequivocal evidence for its performance for dissociation. The authors state that this is demonstrated in the references, however, I could find that only ref 50 has to do with hydrogen dissociation, and it does not look conclusive as benchmarking ROKS was not the focus. Therefore, it would be important to see at least energy profiles of hydrogen abstraction from a water molecule or a water dimer to be able to assess the performance of ROKS.

We thank the Reviewer for highlighting this important aspect regarding the validation of our ROKS approach. We fully agree that a proper assessment of its accuracy is essential. For this reason, we have included in the revised manuscript a new section titled *“Evaluating the Accuracy of ROKS for the Photochemistry of Liquid Water”* (pages 23-26 in the revised version and also reproduced at the beginning of this response letter). In this section, we validate our ROKS results against TDDFT for both mechanisms identified in the photochemistry of liquid water (HAT and PCET), where in both scenarios, we have a bond breaking of water molecule. To gain deeper insight into the performance of ROKS, we additionally tested the method using a hydrogen-bonded water dimer in vacuum, a model that exhibits only the HAT mechanism. We then compared the ROKS and TDDFT results with multiconfigurational CASPT2 calculations that have been previously reported[3]. Although some quantitative differences were observed among the methods, ROKS reproduces the dissociative character predicted by CASPT2 much more accurately than TDDFT. We also validated the ROKS approach using a different exchange–correlation functional and basis set. These additional results are provided in Section *S9 Validation of functional and basis set in ROKS approach* of the Supplementary Information.

Overall, these results confirm that ROKS provides a reliable and computationally efficient framework for describing the key excited-state processes involved in the pho-

tochemistry of liquid water.

2.2: "Later" stages of the trajectories considered here show that S_0 and S_1 states become close in energy for both processes, these stretches of dynamics being relatively long for PCET. It implies that non-adiabatic effects must play considerable role here. I won't suggest to remake everything with surface-hopping approaches, but it's an important issue. Actually computation of non-adiabatic couplings in CP2K is now possible due to the work of A. Hehn et al.

We sincerely thank the Reviewer for highlighting this excellent and important point. We fully agree that non-adiabatic effects can play a significant role in the photochemistry of water. However, several considerations must be clarified to explain why we did not explicitly include non-adiabatic coupling vectors (NACVs) in our present simulations.

First, our study employs the ROKS approach, chosen based on previous works showing that the hydrated electron forms upon excitation through charge transfer to the solvent. In the revised manuscript, we have added a detailed validation section demonstrating that ROKS provides a reliable and physically consistent description of this process (see also **point 2.1**). Unfortunately, at present, CP2K does not include an implementation for calculating NACVs within the ROKS framework, which prevents a direct evaluation of these quantities in our simulations.

Alternatively, we could have performed the simulations using TDDFT. However, it is well known that linear-response TDDFT struggles to accurately describe conical intersections, often predicting incorrect dimensionality between the excited and ground electronic states[7, 8]. One of the most robust and widely used strategies to model non-adiabatic transitions in such cases is to employ an energy gap criterion[7, 8, 9, 10, 11], as we have done in this work. This approach is justified by the mathematical relationship between NACVs and energy gap[12], which shows that NACVs increase sharply when the energy gap between states decreases, effectively capturing the region where non-adiabatic transitions are most likely to occur.

Another possible approach for modeling nonadiabatic dynamics is to use surface hopping methods based on the Landau-Zener (LZ) or Baeck-An (BA) approaches. These algorithms do not explicitly compute NACVs but incorporate their effects implicitly through the dependence of transition probabilities on the energy gap and the curvature of the potential energy surfaces involved in the transition.

We would like to thank the reviewer for bringing to our attention the recent work of A. Hehn et al.[6], which indeed represents an important step toward including non-adiabatic effects with CP2K. However, NACVs between the S_1 and S_0 states were not

computed directly in that work for the same reasons discussed above in the context of TDDFT. Instead, the authors introduced the transition to the ground state via the BA algorithm. Their results showed that the BA method tends to overestimate the transition probability to S_0 (see Figure 6 in Ref.[6]), highlighting the current limitations of this approach.

Considering these challenges, we hope the reviewer understands the practical and methodological reasons that prevented us from including explicit NACVs in our present study. We also hope that our work motivates further development of NACV calculations within the ROKS framework, which would enable more accurate non-adiabatic simulations in the future.

Finally, to provide a more quantitative connection with nonadiabatic dynamics, we have now computed the LZ transition probabilities (P_{LZ}), which have been shown to reproduce the results of traditional surface hopping algorithms that explicitly include NACVs across a variety of systems and electronic structure methods[13, 14, 15, 16]. The new results are presented in the Section *S10 Incorporating Non-Adiabatic Effects into the Photochemistry of Liquid Water* in the Supplementary Information (here we show the main results). As shown in the Figure below, the P_{LZ} values increase significantly near the S_1/S_0 crossing points for representative trajectories of both mechanisms, supporting our choice of using the energy gap criterion to identify non-radiative transitions.

We also add the following in the revised manuscript in the sections (page 31, lines

590-594):

..., justifying the omission of explicitly determining NACVs in our simulations^{113,114}. *To further support this approximation, we evaluated the Landau-Zener (LZ) transition probabilities¹¹⁵, which in addition to the energy gap criterion, also account for the curvature of the potential energy surfaces (see section S10 in the Supplementary Information). This approach has been shown to reproduce the results consistent with those obtained using explicit NACVs in non-adiabatic dynamics^{115–119}*

2.3: Spin densities in ROKS are not clearly defined: CP2K writes .cube files but issues a warning in the output. And indeed, the total multiplicity is singlet and strictly speaking there is no spin density despite the presence of unpaired electrons (AKA singlet diradicals). Please, clarify the point.

We thank the reviewer for raising this clarification. Indeed, in the ROKS formalism the total spin multiplicity is singlet, and therefore the spin density is not rigorously defined. The “spin density” cube files in this case represents the difference in electron distribution between the two open-shell orbitals involved in the excitation. We use these files to perform all the analysis in the present work.

No warnings were printed in the output file, here we show the lines printed in the output in this regard, we also provide inputs to reproduce these results.

The sum of alpha and beta density is written in cube file format to the file:

```
cp2k-roks-dens-ELECTRON_DENSITY-1_0.cube
```

The spin density is written in cube file format to the file:

```
cp2k-roks-dens-SPIN_DENSITY-1_0.cube
```

2.4: It may be instructive to compare the role of solvent dynamics in solvated electron’s formation with the role of solvent dynamics in its reaction with CO2. It appears that there may be useful analogies and generalizations. See:

<https://pubs.acs.org/doi/abs/10.1021/acs.jpcc.0c07859> and

<https://pubs.acs.org/doi/10.1021/acs.jpcc.3c06935>

We thank the reviewer for bringing these papers to our attention. We have now added a discussion in the section where we talk about the coupled rotational and

translational solvent motions that stabilize the hydrated electron (page 18, lines 340-344).

...consistent with experimental findings from ionization studies in liquid water³⁵.

A similar reorganizational response has been reported in ab initio molecular dynamics simulations of carbon dioxide reduction in water^{77,78}, where the breaking of hydrogen bonds in the second solvation shell triggers the stabilization of the excess electron on the carbon dioxide molecule. This process also involves coupled translational and rotational motions of water molecules, closely paralleling the behavior observed in our simulations.

Responses to Reviewer #3:

3.1: The accuracy of the ROKS results depends heavily on the functional used, basis set, and how well the method captures dynamic correlation. A more thorough assessment of the performance for excited-state dynamics in photoionized water, would be valuable.

We thank the Reviewer for highlighting this important point, which was indeed not sufficiently addressed in the original version of our manuscript. In the revised version, we have now included a comprehensive assessment of the reliability of the ROKS approach. In particular, we examined the influence of the exchange-correlation functional and basis set by performing additional calculations using the CAM-B3LYP functional and the TZV2P basis set. The results, now included in Section *S9 Validation of functional and basis set in ROKS approach* in the Supplementary Information, confirm that the ROKS method yields consistent energy gaps and reproduces the same photochemical behavior for both mechanisms (HAT and PCET), see Figure below.

Figure S10: Basis set and exchange-correlation functional validation within ROKS framework. Comparison between the DZVP (green line) and TZV2P (red line) basis sets using the PBEh(40)-rVV10 functional (PBEh) is shown in **Panel A** and **Panel B** for the HAT and PCET mechanisms, respectively. Similarly, comparison between the PBEh (green line) and CAM-B3LYP (CAM, red line) functionals using the DZVP basis set is shown in **Panel C** and **Panel D** for the HAT and PCET mechanisms, respectively. For the PCET trajectory, the PBEh-TZV2P (red line in **Panel B**) and CAM-DZVP (red line, **Panel D**) calculations were performed at configurations sampled every 5 fs to reduce the computational cost.

Furthermore, we have added a new section discussing these validation results in detail to provide a clearer understanding of the robustness and reliability of the ROKS approach (pages 23-26 of the revised manuscript and the beginning of the response letter). The following modification has been made in this regard (page 23, lines

413-418):

...In order to first assess the robustness of ROKS with respect to the choice of exchange–correlation functional and basis set, we performed additional calculations using the CAM-B3LYP functional⁸⁴ and the TZV2P basis set⁸⁵. The results, presented in section S9 in the Supplementary Information, demonstrate that the ROKS method provides consistent energy gaps and reproduces the same behavior for both photochemical pathways (HAT and PCET), confirming its reliability across different electronic structure settings.

3.2: I would appreciate a more detailed description how exactly the ROKS method is used for excited state dynamics. Comparison to other methods such as TDDFT would be valuable.

We thank the Reviewer for highlighting this important point, which was insufficiently addressed in the original version of the manuscript. To clarify the theoretical foundations of the ROKS method, we have added a brief explanation in Section 5.2, along with relevant references for readers interested in further technical details. The corresponding revisions have been incorporated into the revised manuscript (page 30, lines 549-557):

The excited state molecular dynamics (ESMD) simulations were initiated from the first electronic excited state (S_1), computed using the Restricted Open-shell Kohn–Sham (ROKS) method^{46,47,103}. ~~ROKS has proven to be a reliable approach for describing dissociative and charge-transfer states in closed-shell systems^{48–50}, processes that play a central role in the photochemistry of water.~~ *This method provides an alternative approach for calculating singlet excited states in closed-shell systems that are described by non-Aufbau electronic configurations. By minimizing the energy expression $E_S = 2E_{mix} - E_T$, where E_S, E_T, E_{mix} correspond to the energies of the singlet, triplet, and mixed states, respectively, the method effectively accounts for a two-determinant nature of the excited state. Consequently, ROKS offers a computationally efficient and reliable framework for describing dissociative and charge-transfer excited states^{48–50,104}.*

Regarding the validation of our approach against TDDFT, we had already included some comparative calculations in the original version of the manuscript. However, since this concern was raised by all Reviewers, we decided to address it in a more comprehensive manner. In the revised manuscript, we have added a new section titled *“Evaluating the Accuracy of ROKS for the Photochemistry of Liquid Water”* (pages

23-26). In this section, we compare our ROKS results with TDDFT calculations using the same exchange-correlation functional and basis set for two representative trajectories corresponding to each mechanism. To further understand the observed differences between the two methods, we also performed calculations on a hydrogen-bonded water dimer in vacuum and compared the results from ROKS and TDDFT with those obtained from the multiconfigurational CASPT2 method. Although some discrepancies were found among the three approaches, ROKS reproduces the photodissociation behavior of water more accurately and qualitatively aligns better with the CASPT2 reference.

3.3: The authors acknowledge the simplification of not explicitly including non-adiabatic couplings. While it may be reasonable for the present system to consider the energy gaps only, a more rigorous justification or exploration of the potential impact of the approximation used on the results would strengthen the argument.

We thank the Reviewer for highlighting this important point. As this issue was also raised by another Reviewer, to avoid repetition we kindly refer the reviewer to our response to **point 2.2**, where we explain in detail why non-adiabatic coupling vectors (NACVs) cannot be explicitly incorporated into our current approach.

Nevertheless, we have validated our methodology by calculating Landau–Zener (LZ) transition probabilities, which account not only for the energy gap but also for the curvature of the potential energy surfaces. This approach has been shown to reproduce results comparable to those obtained with traditional surface-hopping algorithms that explicitly include NACVs, across a wide range of systems and electronic structure methods[13, 14, 15, 16]. The corresponding results are now presented and discussed in section S10 in of the Supplementary Information. The main result is shown below.

Figure S11: Landau-Zener (LZ) Probabilities. Energy gap (purple line) and LZ probabilities (red line) as a function of time for representative HAT (**Panel A**) and PCET (**Panel B**) trajectories. **Panel C**: shows a 2D density plot of the LZ probabilities versus energy gap for all trajectories, with the inset providing a zoomed-in view at small LZ values. Red points indicate the values at the $S_1 \rightarrow S_0$ crossing events.

We have also included in the revised manuscript the following modification (page 31, lines 590-594):

..., justifying the omission of explicitly determining NACVs in our simulations^{113,114}. *To further support this approximation, we evaluated the Landau-Zener (LZ) transition probabilities¹¹⁵, which in addition to the energy gap criterion, also account for the curvature of the potential energy surfaces (see section S10 in the Supplementary Information). This approach has been shown to reproduce the results consistent with those obtained using explicit NACVs in non-adiabatic dynamics^{115–119}*

As we can see in the Figure, LZ probabilities increase sharply when the energy gap becomes small, confirming the consistency and validity of our energy gap based criterion for both mechanisms.

3.4: Given the computational demands of the method, it is understandable that the authors used a relatively small unit cell with limited sampling of initial configurations. Nevertheless, at least some discussion on the size and sampling effects would be appropriate.

We sincerely thank the Reviewer for raising this important point. The choice of system size was primarily dictated by the high computational cost of our approach. The present unit cell represents the largest system that can be feasibly simulated

within the ROKS framework at the ab initio level used in this study. Importantly, this cell size is consistent with those typically employed in *ab-initio* molecular dynamics studies of bulk water, particularly in investigations of the hydrated electron.

We acknowledge that finite size effects inevitably impose certain limitations. In particular, we have already discussed how the restricted box size affects the degree of electron delocalization upon excitation. This limitation was explicitly noted in the manuscript (page 15, lines 285–288):

Experimental studies have reported initial gyration radius of around 20-40 Å^{34,67} immediately after photoabsorption. Although our method captures the same qualitative behavior, direct comparison with experiment is limited by the size of the simulation box (12.42 Å), which restricts the maximum observable extent of delocalization⁷³.

Regarding statistical sampling, our study includes 100 independent trajectories, which we believe is sufficient to capture representative trends and provide meaningful ensemble averaged insights. Nevertheless, we acknowledge that increasing the sampling size could reveal additional rare events or subtle mechanistic features.

To address both the system size and sampling limitations, we have expanded the discussion in the final part of the *Conclusions* section, emphasizing the potential of machine learning approaches to overcome these challenges (page 28, lines 521-523):

On the more methodological side, it is interesting to explore whether our simulations can be used to develop machine-learning approaches^{90–94} to examine the excited-state dynamics of aqueous systems ~~which would allow for expanding the sampling and statistics. This is an area we are actively pursuing and will be the subject of a future study.~~ *Such methods would enable simulations with larger unit cells and a greater number of trajectories, potentially improving agreement with experimental observations and revealing new physical insights.*

3.5: I am puzzled by the author’s finding of the H3O radical. This must be a very unstable species (and the authors indeed discuss its transient character) but I wonder if it is not more like a H3O+...e- contact “ion pair”.

We thank the Reviewer for raising this important point. Our analysis reveals that within the HAT mechanism, the transient H₃O• species indeed forms, as the excess electron density is localized near one of the hydrogens bonded to oxygen, an aspect clarified in the *Analysis Protocols* section. We did not find the formation of H₃O⁺

$-e^-$ pair within the HAT mechanism. In contrast, the PCET mechanism leads to the formation of H_3O^+ , HO^\bullet , and a solvated electron. In this case a $\text{H}_3\text{O}^+ - e^-$ pair emerges, mediated by surrounding water molecules that stabilize both species, as illustrated in Figure 2. We provide cube files for both mechanism in the Zenodo repository.

In the present work, we chose to focus on the $\text{H}_3\text{O}^\bullet$ radical because it has not yet been experimentally detected, and thus providing insights into this species. We also focus on the $\text{HO}^\bullet - \text{H}_3\text{O}^+$ pair, since it has been experimentally observed, allowing us to better validate our computational approach. Nonetheless, the behavior and dynamics of the $\text{H}_3\text{O}^+ - e^-$ represent an intriguing topic that warrants further investigation in future studies.

Responses to Reviewer #4:

4.1: Their key analysis tool is to use “dilution factor” or “inverse participation ratio” to quantify how many water molecules an excess electron is delocalized over. Note: strictly speaking the “PDFs” in Fig. 1BC are counts, not probability distribution functions; they should be normalized, or maybe called probability counts.

We thank the reviewer for bringing this error to our attention. The reviewer is indeed correct, we used the IPR because it provides a continuous measure that reflects the number of water molecules involved in the excitation. We have now corrected Figures 1B and 1C, changing the label from *PDF* to *Probability counts*.

4.2: The conclusion is almost complete localization: the peak in Fig. 1B for 1 water is 50, at 2 water molecules we are already down to ~ 2 counts, or 4%. This need to be mentioned more clearly So, the further analysis does not change the view compare to with one water. The figures in Fig.2 B could clearer, they are hard to see on the screen.

We thank the Reviewer for raising this concern. We would like to clarify that the IPR values shown on the x-axis of Figure 1B are continuous variables that represent the number of water molecules participating in the excitation. Therefore, in order to explain the data point corresponding to “2” on the x-axis we should look at all the configurations with IPR values near 2, which are more than 4% of the initial conditions. This point has now been clarified in the Analysis Protocol section (page 34, line 640):

This metric *provides a continuous measure* of how many water molecules participate in the excitation.

We agree with the Reviewer that most excitations primarily involve a single water molecule. Moreover, within the ensemble of simulations performed in this work, no significant differences in the photochemical behavior were observed as a function of the number of water molecules involved in the excitation.

In the revised version of our manuscript, Figure 2 has been modified to enhance its clarity and readability.

4.3: Defects account for 3/4 of the binding sites, mainly missing acceptor sites: a missing acceptor bond means the water molecule has a lone pair on oxygen not bonded to an H from another water molecule, destabilizing the n to sigma* transition to 8 eV. What is missing here is a further aspect: the steric room available next to an unsatisfied acceptor bond also helps the HO + H dissociation process by making room for the recoiling OH and H.

We thank the Reviewer for raising this important point, which was indeed missing from our original manuscript. Since this comment is closely related to a concern raised by another Reviewer (see **point 1.4**), we have addressed both points together. The following modifications have been included in the revised manuscript in Section 2.3 (pages 14–15, lines 264–274):

... An interesting observation is that approximately 20% of the trajectories follow the mechanism involving the formation of a hydrogen atom in an empty cavity (Figure 3B) with a lifetime of 13 fs, while the remaining 33% correspond to the alternative HAT pathway (Figures 3C and 3D) exhibiting a longer lifetime of 34 fs. These results correlate well with the defect analysis presented in Figure 1C. When the dissociating water molecule has a missing hydrogen-bond donor (AD or AAD defect), there is sufficient space available to allow an ultrafast hydrogen atom dissociation into a cavity (~13 fs), in excellent agreement with values reported for a hydrogen-bonded water dimer in vacuum^{71,72}. In contrast, when the dissociating molecule lacks a hydrogen-bond acceptor (ADD defect), the surrounding hydrogen-bond network in the condensed phase restricts this motion, delaying the non-radiative decay and promoting the formation of the transient H₃O[•] species.

4.4: The definition of the decay lifetime on p. 9 is somewhat arbitrary but consistent, sufficient to distinguish processes differing by more than a

factor of 10 in lifetime. May be this definition can be made more precise.

We thank the reviewer for bringing this important point to our attention. In response, we have revised the *Analysis Protocols* section to include explicit definitions of the lifetimes and quantum yields (page 35-36, lines 680–688).

Based on the preceding analysis, each trajectory can be unambiguously assigned to a specific mechanism. This allows us to determine the quantum yield and lifetime associated with each mechanism using the following expression:

$$QY_M = \frac{\# \text{Trajectories with } M}{\# \text{Total trajectories}} * 100 \quad (6)$$

$$\tau_M = \frac{\sum_i^{N_M} t_i^M}{\# \text{Total trajectories with } M} \quad (7)$$

where M represent the mechanism (HAT or PCET), QY, τ are the quantum yields and lifetimes, N_M is the number of trajectories with the M mechanism and t_i^M is the time at which the trajectory i with the M mechanism finds the "crossing point".

4.5: In the PCET mechanism ($n \rightarrow \sigma^*$ H₂O + H₂O \rightarrow HO + e⁻ + H⁺ + H₂O \rightarrow HO + e⁻ + H₃O⁺), the e⁻ quickly becomes localized in a trapped state. The OH and H₃O⁺ become quite well separated (6 Å in Fig. 5D). Their theoretical study reveals three distinct minima in the HO* and (H₃O⁺) distance which are in nice agreement with the study of the recent electron diffraction experiments upon photoionization. These could be rationalized by the formation of contact ion radical pair and the separation mediated by the solvent. The results are not surprising, the new aspect is that the entire process can be modelled from photoexcitation up to the localized electron directly capturing the solvent dynamics.

We thank the Reviewer for their comment and for emphasizing the main findings of our study. We believe that the present work represents a significant step forward in understanding the complex photochemistry of liquid water. For the first time, we demonstrate how all the key processes: including water dissociation, formation of hydroxyl and hydronium radicals, electron generation, hydronium ion and collective water motions evolve within a unified framework. Our results not only reproduce these intricate phenomena starting from neutral liquid water but also provide new physical insights, particularly into the very early stages of the photochemical process and the early birth of the hydrated electron. This initial regime has been largely unexplored

in previous theoretical studies, and we hope that our findings will stimulate further investigations under a broader range of aqueous conditions.

Responses to Reviewer #5:

5.1: I co-reviewed this manuscript with one of the reviewers who provided the listed reports. This is part of the Nature Communications initiative to facilitate training in peer review and to provide appropriate recognition for Early Career Researchers who co-review manuscripts.

We sincerely thank the Reviewer for their time and effort in evaluating our work. We truly appreciate all the feedback which helped us to improve the quality and clarity of our manuscript.

References

- [1] Takeshi Yanai, David P Tew, and Nicholas C Handy. A new hybrid exchange–correlation functional using the coulomb-attenuating method (cam-b3lyp). Chemical physics letters, 393(1-3):51–57, 2004.
- [2] Joost VandeVondele and Jürg Hutter. Gaussian basis sets for accurate calculations on molecular systems in gas and condensed phases. The Journal of chemical physics, 127(11), 2007.
- [3] Javier Segarra-Marti, Daniel Roca-Sanjuan, Manuela Merchan, and Roland Lindh. On the photophysics and photochemistry of the water dimer. The Journal of Chemical Physics, 137(24), 2012.
- [4] Anupriya Kumar, Maciej Kołaski, Han Myoung Lee, and Kwang S Kim. Photoexcitation and photoionization dynamics of water photolysis. The Journal of Physical Chemistry A, 112(24):5502–5508, 2008.
- [5] Dario Rocca, Ralph Gebauer, Yousef Saad, and Stefano Baroni. Turbo charging time-dependent density-functional theory with lanczos chains. The Journal of Chemical Physics, 128(15), 2008.
- [6] Jan-Robert Vogt, Michael Schulz, Rafael Souza Mattos, Mario Barbatti, Maurizio Persico, Giovanni Granucci, Jurg Hutter, and Anna Hehn. A density functional theory and semiempirical framework for trajectory surface hopping on extended systems. Journal of Chemical Theory and Computation, 2025.
- [7] Rachel Crespo-Otero and Mario Barbatti. Recent advances and perspectives on nonadiabatic mixed quantum–classical dynamics. Chemical reviews, 118(15):7026–7068, 2018.
- [8] Antonio Prlj, Jack T Taylor, Jiří Janoš, Petr Slavíček, Federica Agostini, and Basile FE Curchod. Best practices for nonadiabatic molecular dynamics simulations. arXiv preprint arXiv:2508.05263, 2025.
- [9] Rachel Crespo-Otero and Mario Barbatti. Cr (co) 6 photochemistry: Semi-classical study of uv absorption spectral intensities and dynamics of photodissociation. The Journal of chemical physics, 134(16), 2011.
- [10] Felix Plasser, Rachel Crespo-Otero, Marek Pederzoli, Jiri Pittner, Hans Lischka, and Mario Barbatti. Surface hopping dynamics with correlated single-reference methods: 9h-adenine as a case study. Journal of chemical theory and computation, 10(4):1395–1405, 2014.

- [11] Lea M Ibele, Pedro A Sanchez-Murcia, Sebastian Mai, Juan J Nogueira, and Leticia González. Excimer intermediates en route to long-lived charge-transfer states in single-stranded adenine dna as revealed by nonadiabatic dynamics. The Journal of Physical Chemistry Letters, 11(18):7483–7488, 2020.
- [12] John M Herbert, Xing Zhang, Adrian F Morrison, and Jie Liu. Beyond time-dependent density functional theory using only single excitations: Methods for computational studies of excited states in complex systems. Accounts of chemical research, 49(5):931–941, 2016.
- [13] Jiri Suchan, Jiri Janoš, and Petr Slavicek. Pragmatic approach to photodynamics: Mixed landau–zener surface hopping with intersystem crossing. Journal of Chemical Theory and Computation, 16(9):5809–5820, 2020.
- [14] Nina Tokić, Tomislav Piteša, Antonio Prlj, Marin Sapunar, and Naa Došlić. Advantages and limitations of landau-zener surface hopping dynamics. Croatica chemica acta, 97(4):P1–P11, 2024.
- [15] Weiwei Xie and Wolfgang Domcke. Accuracy of trajectory surface-hopping methods: Test for a two-dimensional model of the photodissociation of phenol. The Journal of Chemical Physics, 147(18), 2017.
- [16] Gonzalo Díaz Mirón, Carlos R Lien-Medrano, Debarshi Banerjee, Uriel N Morzan, Michael A Sentef, Ralph Gebauer, and Ali Hassanali. Exploring the mechanisms behind non-aromatic fluorescence with the density functional tight binding method. Journal of Chemical Theory and Computation, 20(9):3864–3878, 2024.

Comments for: Simulating the Photochemical Birth of the Hydrated Electron in Liquid Water

January 2026

Dear Reviewers,

We sincerely appreciate the time and effort that the Reviewers have dedicated to evaluating our manuscript and providing thoughtful and constructive feedback. In the following sections, we provide detailed responses to each comment raised by the Reviewers. The corresponding revisions are indicated in the file “manuscript_marked”, where page and line numbers are referenced for clarity.

Responses to Reviewer #1:

1.1: I thank the authors for thoroughly addressing my comments. I recommend publication of the manuscript in Nature Communications. In addition, it would be helpful if the authors could clarify the following point: In Figures 1B and 1C, the y-axis is labeled ”Probability counts”, which is not normalized and therefore does not sum to 1. However, the caption states that these panels show the probability distribution function, which implies a normalized quantity. The figure would be easier to interpret if this inconsistency were resolved.

We sincerely thank the Reviewer for this comment and for recommending our manuscript for publication. We apologize for this oversight and have now addressed the issue in the caption of Figure 1. The following modification has been made in the revised manuscript (page 8, Figure 1):

... **Panel B:** Probability ~~Distribution Function (PDF)~~ *Counts* of water molecules involved in the excitation using the Inverse Participation Ratio of the spin densities for all the conformations. The insets show two examples involving 1 and 5 water molecules. **Panel C:** ~~PDF~~ *Probability Counts* of the different defects in the Hydrogen Bond (HB) Network of all the water molecules involved in the initial excitation. ...

Responses to Reviewer #2:

2.1: In the revised version of the manuscript "The Photochemical Birth of the Hydrated Electron in Liquid Water" the authors have addressed main issues pointed out in the first round of peer review: provided validation of ROKS for the HAT mechanism, discussed and evaluated the importance non-adiabatic effects and performed simple non-adiabatic (Landau-Zener theory). In addition, they referred to more relevant publications. I will support the publication of this work in Nature Communications if the discussion of the benchmark of ROKS against CASPT2 is done properly and fairly: their evaluation of the methods is way too optimistic, contradicting the data. Unfortunately, I have reasons to believe it is not the case. Panels D and E in Figure 7 show what I was afraid such a benchmark would show: very significant qualitative differences. Yes, the barrier height is indeed similar, but the geometry of TS is embarrassingly different (which is ingeniously, and reasonably, downplayed). However, the values in between the reactant and the conical intersection are nowhere near. Say at the coordinate value of 2 (units are not given at the axis, please add it) the the CASPT2 energy is ca. 2 times larger than that of ROKS, the difference being immense 2 eV. The shape of conical intersection region is dramatically different: it is clear from the figure that non-adiabatic effects at the multireference level of theory will be significantly larger since the gap squared appears in the exponential in the Landau-Zener theory. I believe it's non proper to write about "close agreement with TDDFT and CASPT2" and so on in the last paragraph of Section 3. Instead, the authors should come up with realistic evaluation of their approach as at most semiquantitative.

We thank the Reviewer for bringing this important point to our attention. We would first like to clarify how the validation against CASPT2 was carried out. In the CASPT2 study by Segarra et al.[1], the authors reported a minimum energy path (MEP) analysis but provided optimized geometries only for the ground state mini-

mum and the conical intersection. Starting from these two geometries, we constructed a linear interpolated path (LIP) and evaluated the ROKS and TDDFT energies along this coordinate. As a consequence, a direct, point-by-point comparison between the published MEP with CASPT2 and LIP pathways (that we construct with ROKS and TDDFT) is not possible. In particular, only the initial geometry shown in Figure 7 panels D-F corresponds to the calculation with the same nuclear configuration at all levels of theory. We recognize that this limitation was not sufficiently clear in the original version of the manuscript, and we have therefore revised the text to explicitly state this point. The corresponding modification is reported below (page 25, lines 458-462):

Using both the ground-state and conical intersection optimized geometries at this high level of theory, we computed the potential energy surfaces along the Linear Interpolated Path (LIP) with ROKS (Figure 7E) and TDDFT (Figure 7F) employing the CAM-B3LYP functional. *It is worth emphasizing that the numerical values along horizontal axes for the MEP and LIP coordinates do not correspond to identical nuclear configurations, except for the initial point shown in Figure 7D-F. We thus focus on the overall trends of ground and excited state potential energy surfaces.*

We agree with the Reviewer that our original statement regarding the quantitative validation was perhaps overly optimistic. In particular, the difference of approximately 0.8 Å in the HO•–H• distance at the conical intersection between ROKS and CASPT2 highlights a significant discrepancy in the predicted intersection topology. To address this point more carefully, we have revised the discussion to better reflect the qualitative rather than quantitative nature of this comparison. The corresponding modification is reported below (pages 25-26, lines 465-470):

~~Although this difference is not insignificant, a 2 Å separation already indicates a fully dissociated configuration, confirming that ROKS and CASPT2 capture the same underlying photodissociation physics~~ *Although an O-H distance of 2 Å could be considered as dissociated geometry, the topology of the conical intersection differs between the two methods. Despite this difference, ROKS reproduces the same essential physical process, namely the hydrogen atom dissociation.*

Finally, we have revised and expanded the concluding discussion of the validation section to provide a more balanced assessment of the strengths and limitations of the ROKS approach. These changes are reported below (pages 26–27, lines 476–494).

Taken all together, these results provide a comprehensive validation of the ROKS approach. We demonstrate that ROKS captures the essential physics of both the HAT and PCET photochemical pathways in *liquid* water, ~~in close agreement with TDDFT and CASPT2 benchmarks. This consistency across different mechanisms and details of the underlying electronic structure supports the robustness of our methodology and reinforces confidence in its predictive capability for modeling the excited-state dynamics of liquid water. Based on the gas phase water dimer model, ROKS does not reproduce the conical intersection geometry for the HAT mechanism. In particular, a difference of approximately 0.8 Å is observed in the HO•-H• distance at the conical intersection when compared with CASPT2. Nevertheless, ROKS qualitatively describes the same photodissociation pathway, reproducing both the slope of the excited-state potential energy surface and the presence of the relevant chemical species along the dissociation coordinate. While such differences may lead to deviations in the predicted HAT lifetimes in vacuum, we expect their impact to be reduced in the condensed phase. Indeed, in bulk liquid water, ROKS yields lifetimes and quantum yields that are in good agreement with experimental observations. Overall, the extensive validation presented here demonstrates the robustness of our methodology and reinforces confidence in its predictive capability for modeling the excited-state dynamics and photochemistry of liquid water. We nonetheless caution that more systematic studies with multireference methods for condensed phase aqueous systems needs to be explored in the future.~~

We believe that these revisions more accurately convey the scope of the validation presented in our manuscript.

Responses to Reviewer #3:

3.1: The authors satisfactorily addressed my comments, so I believe the ms. may be published.

We sincerely thank the Reviewer for their positive assessment of our responses and for supporting the publication of our manuscript.

Responses to Reviewer #4:

4.1: This is a theoretical simulation study on the birth of a solvated electron following all the steps thorough from the photoexcitation ,i.e. from

the ground state to the excited state and then the formation of the hydrated electron up to localization including radical formation. New experimental techniques have been developed including e.g. ultrafast electron diffraction which allows to monitor this process experimentally. Thus a theoretical study which involves all steps is of relevance and has not been reported before. The data analysis and the choice of the theoretical methods should be checked in detail by theoretical referees. Being outside of this field I can only state that the results are interesting and noteworthy.

We sincerely thank the Reviewer for their thoughtful and positive assessment of our work. We believe that our study provides complementary insight into these experiments by offering a molecular level interpretation of the processes involved in the birth and localization of the hydrated electron, which we hope will be useful for guiding and interpreting future experimental investigations.

Responses to Reviewer #5:

5.1: I co-reviewed this manuscript with one of the reviewers who provided the listed reports. This is part of the Nature Communications initiative to facilitate training in peer review and to provide appropriate recognition for Early Career Researchers who co-review manuscripts.

We sincerely thank the Reviewer for their time and effort in evaluating our work. We truly appreciate all the feedback which helped us to improve the quality and clarity of our manuscript.

References

- [1] Javier Segarra-Martí, Daniel Roca-Sanjuan, Manuela Merchán, and Roland Lindh. On the photophysics and photochemistry of the water dimer. The Journal of Chemical Physics, 137(24), 2012.